# Factors Influencing the Degree of Employee Involvement in Preventive Nutrition and Physical Activity Web-Based Programs in Medium and Small Enterprises

**DOI:** 10.3390/nu15245129

**Published:** 2023-12-17

**Authors:** Mojca Stubelj, Klemen Širok, Suzana Laporšek, Simona Perčič

**Affiliations:** 1Faculty of Health Sciences, University of Primorska, 6310 Izola, Slovenia; mojca.stubelj@fvz.upr.si (M.S.); klemen.sirok@fvz.upr.si (K.Š.); 2Faculty of Management, University of Primorska, 6000 Koper, Slovenia; suzana.laporsek@fm-kp.si; 3National Institute of Public Health Ljubljana, 1000 Ljubljana, Slovenia

**Keywords:** diet, food, nutrition, health promotion, physical activity, workplace, obesity, overweight, nutrition campaign

## Abstract

This cross-sectional study was part of a comprehensive workplace health promotion program (WHPP) implemented in the work environment of small- and medium-sized enterprises (SMEs) in Western Slovenia, which included web-based educational campaigns aimed at promoting positive lifestyle changes among workers, including healthy eating habits and physical activity. As part of this program, which included campaigns in the areas of stress management, ergonomics, sleep hygiene, communication, work climate, and absenteeism, we developed and examined the engagement frequency in web-based content on healthy eating and physical activity for the companies included in the WHPP. This part of the project lasted from 2020 to 2022. Prior to the educational campaign, participants voluntarily completed a screening questionnaire. We analyzed the patterns of 370 workers in terms of their job classification (predominantly sedentary, predominantly standing, or predominantly physical), body mass index (BMI), gender, age, and selected indicators of diet and physical activity. Of the 88 companies participating in the WHPP, 26 took part in our web-based educational campaigns on nutrition and physical activity. Through an empirical analysis using descriptive and inferential statistics and a linear regression, we found that, on average, the engagement frequency (i.e., proportion of educational personal e-mails read) was highest among men with sedentary work, with older employees participating more actively than expected. Moreover, workers with good dietary habits and a favorable BMI proved to be the most avid readers of the web-based campaigns. Despite the overall low participant engagement frequency, it is clear that web-based educational campaigns are more appealing to workers with sedentary jobs and good dietary habits. This study provides valuable information on the potential effectiveness of appropriate workplace health promotion campaigns for SMEs and public health practices.

## 1. Introduction

The workplace, where the working population spends a large part of the day, serves as a compelling focal point for improving workers’ habitual behaviors. The implementation of various intervention strategies has generally proven useful in combating overweight and obesity [1,2,3]. Obesity, along with low physical activity, is the most common cause of chronic degenerative non-communicable diseases (NCDs), such as type 2 diabetes, cardiovascular disease, hypertension and stroke, various forms of cancer, and mental disorders [4]. Obesity and obesity-related diseases are associated with an increased risk of sick leave and permanent absence from work [5]. Conversely, educational interventions within workplace health promotion programs (WHPPs) for weight loss can lead to improvements in workers’ metabolic parameters [6], cardiometabolic risk factors [7], and the prevalence of metabolic syndrome [8]. Moreover, such interventions can increase work productivity, reduce the costs of illness-related absenteeism, and improve workers’ general health [9]. A well-designed intervention within a WHPP is crucial to instigate changes in dietary and physical activity habits [10]. Web-based training is also an effective tool for motivating individuals to change their poor dietary habits [11,12], despite this impact being relatively modest [13,14]. Nevertheless, the advantages of web-based educational campaigns on nutrition are numerous: they reach a larger groups of people with minimal effort on the part of dietitians, they are flexible in terms of time, and they are cost-effective. Web-based nutrition information has demonstrated favorable results in educating low-income individuals [15]. It has also been observed that more and more people are turning to the internet for information on diets and nutrition to combat obesity-related problems [16]. In this context, web-based campaigns sent via e-mail show promise in promoting physical activity and nutrition in the workplace [17]. To reach a large number of employees, it is therefore reasonable to generate email-delivered web-based content that is credible and consistent with national public health policies. 

Most research on workplace interventions has been conducted on middle-aged populations in large corporations and has lacked data on specific physical demands of the workplace or the extent to which the distributed web-based information was read (i.e., the participants’ engagement frequency). Furthermore, small- and medium-sized enterprises (SMEs) tend to be associated with poorer occupational health and safety outcomes compared to large enterprises due to a lack of knowledge, time, motivation, resources, and skills [18]. Whilst SMEs struggle with a range of complex requirements to manage the health and well-being of their employees, these work environments tend to be harder to reach and more difficult to change [19]. As there appears to be a lack of research in this area, our study focused precisely on SMEs. Our aim was to explore whether there are differences in the extent to which employees of different occupations, body mass index (BMI) values, genders, ages, and selected indicators of dietary and activity habits read the information delivered to their personal e-mails. The results would be of interest to public health planners and would provide insight into previously overlooked data.

Accordingly, the primary objective of this study was to identify the differences in the engagement frequency in e-mail educational campaigns between employees performing different types of work classified as (a) predominantly sedentary, (b) predominantly standing, and (c) predominantly physical, with the additional aim of analyzing these differences by BMI, gender, age, and selected indicators of dietary and activity habits in SMEs. 

## 2. Materials and Methods

### 2.1. Study Design

A comprehensive WHPP was conducted in work environments of SMEs in Western Slovenia. The entire project started on 1 September 2017 and was completed on 30 September 2022. A detailed explanation of the project is available online [20]. This section provides a brief explanation of a part of the project, namely the campaign in the field of nutrition and physical activity, which lasted from 2020 to 2022.

All employees participating in the study were ensured anonymity and confidentiality in accordance with the European Union General Data Protection Regulation (GDPR). They were also assured that the data collected from employees would be treated confidentially and would not be shared. Participants were informed of their right to withdraw their consent to participate at any time. Both employers and employees were informed that the data collected would be used for research purposes as part of the study and that access to the information collected would be restricted to the research team.

The participation of all employees of the selected SMEs was voluntary. Initial contact with the company managers was established through telephone calls and/or visits with a brief explanation of the program. The screening questionnaire was accessible online via an e-platform (see below), with a unique ID assigned to each employee and/or employer to ensure data anonymity. By completing the screening questionnaire, each participant was automatically registered as a new user on the e-platform. The responses to the screening questionnaires were then analyzed for each individual company and the aggregated results were presented to each employer. By June 2019, a total of 1061 employees had completed the screening questionnaire. Of these, 370 took part in the nutrition and physical activity campaigns.

### 2.2. E-Platform

The e-platform used for the purposes of the study is a web-based solution aimed specifically at employees, health promoters, and managers, and is based on a CRM (customer relationship management) system. The platform offers over 250 landing pages with video and other educational content, templates, and interactive questionnaires, as well as active lifestyle challenges that assess participants’ activities and also reward them for certain activities. By using the e-platform, companies and employees are encouraged to change their health-related habits in the work environment through targeted activity programs or, in CRM terms, health promotion campaigns. While these campaigns were primarily distributed via personal e-mails, the e-platform also incorporates short message service (SMS) as a secondary communication channel. The e-platform serves several functions and provides the following: (a) a screening questionnaire with participant registration function (as described above); (b) e-mail distribution of individual health campaigns; (c) e-mail distribution of general WHPP information to companies; (d) generation and e-mail distribution of interactive content such as screening questionnaires and active lifestyle challenges; and (e) user activity monitoring, in-depth user analysis, and user content evaluation. 

The CRM-based e-platform facilitates user activity monitoring and the automatic sending of reminders in case of participant inactivity through an integrated e-mail tracking function (i.e., e-mail delivery monitoring). Each e-mail sent via the e-platform is monitored in terms of the number of users to whom it was sent, the number of users to whom it could not be delivered (i.e., the email remained unopened), as well as the number of users who read the e-mail. Each email sent during the study contained a link to content on a landing page, which, in turn, included further links to content or video. At the conclusion of each landing page, participants were prompted to provide a content rating. We established that an email was ‘read’ only for those individuals who actively provided a content rating. The e-platform also keeps track of the most frequently read emails. Combining these two data sources on the reach of the content allows for an in-depth analysis of participant activity and the quality and usefulness of the content. 

To further encourage participation, companies were offered active lifestyle challenges. Employees were incentivized with small rewards depending on the number of challenges they completed in the areas of physical activity, nutrition, sleep, and interpersonal relationships. For daily reminders about healthy lifestyle, we designed posters for the companies to display in high-traffic areas, such as canteens and corridors.

### 2.3. Screening Questionnaire 

The screening questionnaire comprised twenty-four questions, eight of which focused on dietary habits, which, depending on the answer, could be deemed unhealthy and are defined as risk factors in Slovenia [21]. The questions were: How many meals do you eat per day? How often do you consume vegetables? How often do you consume fruit? How often do you consume red meat? How often do you consume fish and fish products? How often do you consume whole grain products/meals (cereals or porridge)? How often do you consume fried foods? How often do you consume beverages with added sugar (carbonated sweet beverages, flavored water, fruit syrup)? Participants’ dietary habits were categorized as predominantly good if they reported five or more healthy dietary habits. 

The screening questionnaire also included questions on physical activity based on the International Physical Activity Questionnaire [22]. According to the WHO [23], we categorized the participants who reported engaging in at least 75 min of vigorous physical activity per week or at least 150 min of moderate physical activity per week as having predominantly good physical activity habits.

Based on the employees’ responses, we categorized their work as (a) predominantly sedentary, (b) predominantly standing, or (c) predominantly physical. In order to categorize three work groups’ survey respondents, we utilized the FAO/WHO/UNU [24] classification and Ministry of Health of the Republic of Slovenia Guidelines for Healthy Eating for Employees [25], which is predicated on the total energy expenditure and physical activity level of adults. This classification system, comprising sedentary or light activity lifestyle, active or moderately active lifestyle, and vigorous or vigorously active lifestyle, was adapted to the context of the working environment. This adaptation is grounded in the hypothesis that distinct types of work are associated with varying energy expenditures, which, in turn, correlate with diverse dietary needs and habits [25]. Additional information collected via the questionnaire included respondents’ height (cm), weight (kg), age (years), and gender. Participants’ BMI was calculated as their weight (kg) divided by their height (m) squared.

### 2.4. Web-Based Educational Campaign 

Web-based educational materials on nutrition were distributed via the CRM system in the form of a health promotion campaign. Each employee received one unit of content per week to their personal e-mail. This content varied in format and included text (e.g., the benefits of a healthy diet), visuals (e.g., the importance of hydration), and videos (e.g., advice on meal or breakfast preparation or exercises for active rest) and was supplemented with hyperlinks to the relevant national recommendations. As part of this health promotion campaign, each employee received a total of 15 content units over a three-month period. The web-based educational units on nutrition covered the following topics: the benefits of a healthy diet, understanding meal rhythms, determining energy requirements, reading food labels, planning breakfast and other meals for work, determining portion sizes, consuming fruits and vegetables, assembling a healthy plate, following the Mediterranean diet, staying hydrated, controlling salt and spice intake, and following the planetary health diet. The e-content in forms’ short text supplemented by links to articles was collaboratively created by a dietitian, a nutritionist, and a physician, providing a comprehensive understanding of the benefits of a healthy diet. Information on topics such as obesity, sugar consumption, benefits of breakfast, vitamins (B and D), healthy foods, nutrition for a healthy heart, colon cancer and nutrition, and safe foods on vacation, was provided by the physician. A poster highlighting the importance of healthy foods was designed by the dietitian and nutritionist. E-materials on food rhythms, including texts and quizzes for awareness, and the determination of one’s energy requirements [25] through personalized calculations were also included. For practical applications, guidelines for planning balanced meals in various situations, offering twenty recipes and video recordings, were provided by the nutritionist. Effective portion size measurement was succinctly described in a short article and on a poster. They also emphasized fruit and vegetable consumption and were accompanied by quick tips on assembling a balanced meal. The importance of adequate fluid intake was also stressed, providing recipes, video recordings for drink preparation, and posters. Addressing salt and spice consumption, information on benefits of spices, presentations of various spices, and ways to reduce salt intake was offered. Content on specific diets, such as the Mediterranean diet, including texts and quizzes, was also prepared by dietitians and nutritionists, as well as the concept of healthy eating for the planet. The web-based educational materials on physical activity covered the following units: Active Rest, Exercise at Home, Workplace Adaptation, 3 min for Health, His Majesty—the Squat, Your Work Exhausts You, How to Straighten Your Posture, and Stand Up Now. All physical activity materials, overseen by an experienced kinesiologist and physiotherapist, featured a text followed by ten videos demonstrating correct exercise execution for various activities in different occupations.

### 2.5. Statistical Analyses

To describe the basic characteristics of the participants, we used descriptive statistics, including mean and standard deviation (SD). We obtained information on the number and type of web-based campaigns received and read by each employee via the e-platform. This allowed us to calculate each participant’s engagement (i.e., the proportion of e-mail campaigns read). We utilized descriptive statistics (mean and SD) to examine the engagement frequency across different groups. To determine statistically significant differences between the groups, we paired descriptive statistics with the results of inferential statistics. Given that the variables were not normally distributed, we applied the Wilcoxon–Mann–Whitney non-parametric test for categories with two independent groups and the Kruskal–Wallis H test for categories with three or more independent groups. To describe distribution of relative engagement frequency with the emailed content across selected demographic groups and/or habits we used Kernel density plots. To determine the differences between age groups, types of work, and pairwise comparisons of BMI distribution across the types of work, we conducted the Dunn’s post hoc test. To examine the relationship between employees’ engagement frequency and their behavioral patterns, we used multiple linear regression, controlling for employee characteristics. For all categorical variables, we used the first group as the baseline group. The following regression model was estimated:$$Y_{i}=\beta_{0}+\beta_{1}\cdot {Age_{group}}_{i}+\beta_{2}\cdot {Diet}_{i}+\beta_{3}\cdot {PA}_{i}+\beta_{1}\cdot {Jo\mathrm{b\_t}ype}_{i}+\varepsilon_{i},$$
where *Y_i_* represents the dependent variable of engagement frequency by individual employee *i*, *Age*_*group_i_* represents the age group in which individual *i* was classified (we had three age groups, 20–29 years, 30–44 years, 45–70 years, and group with missing data on age), *Diet_i_* represents the dietary habits of individual employees, *PA_i_* represents physical activity habits, *Job*_*type_i_* is type of work the employee performed (sedentary, standing, or physical work). The regression coefficient $\beta_{0}$ represents the regression constant, whereas $\varepsilon_{i}$ is the error term.

All statistical analyses were conducted using the STATA version 14 statistical software package. The study protocol was approved by the National Medical Ethics Committee of Slovenia (No. 0120–345/2018/9).

## 3. Results

### 3.1. Employee Characteristics

Of the 370 employees who participated in the study, 46.5% were women and 53.5% were men. Table 1 shows the main characteristics of the sample, stratified by gender. The majority of employees were in the 45–70 age group. Just over 57% had sedentary jobs. The percentage of female employees of normal weight was 38.4%, and the percentage of male employees of normal weight was 16.7%. A significant proportion of participants (37.6%) of both genders did not report their weight and height information, preventing us from calculating their BMI. The indicators of unhealthy dietary habits were highest for fish consumption and lowest for sugary beverage consumption. Overall, a higher proportion of women (67.4%) reported predominantly good dietary habits (i.e., having five or more good dietary habits) compared to men (50%). There were no discernible gender differences in physical activity.

### 3.2. Engagement in Web-Based Campaigns 

As described in Section 2.4, all employees who participated were sent the web-based health promotion campaign by email. Using the data from the e-platform, we identified the employees who had received the web-based campaign and those who had actually engaged with it by reading its content. Based on this, we calculated the proportion of emails read (engagement frequency) at the individual employee level. 

Table 2 presents descriptive statistics on engagement frequency by demographic group, job type, dietary habits, and physical activity. These descriptive results were supplemented by the results of inferential statistics. The figures in Table 2 illustrate the distribution of the participants’ relative engagement frequency with the email content across selected demographic groups and/or habits.

As shown in Table 2, the engagement frequency was higher among men; however, this difference in means was not statistically significant. In terms of age demographics, older employees recorded a substantially higher engagement frequency than younger employees, with a difference in means exceeding 10 percentage points. As can be seen in Figure 1, this trend can also be observed in the distribution of engagement frequencies across age groups. The results confirmed that there was a statistically significant difference in engagement frequency across the age groups overall (at *p* < 0.05). Statistically significant differences in engagement frequency were between the 45–70 age group and the 30–44 age group (*p* = 0.013) and between the 45–70 and 20–29 age groups (*p* = 0.008). No statistically significant differences were found between the youngest and the middle age group. 

With regard to dietary habits, the results show that the engagement frequency of the employees with poor dietary habits was on average 6.6 percentage points lower than that of the employees with good dietary habits, with a statistically significant difference in means (*p* < 0.05). A further differentiation by gender (see Figure 2) shows that both women and men with poor dietary habits had a lower engagement frequency (on average, 28.9% and 35.6%, respectively) than employees with good dietary habits. Interestingly, men with good dietary habits had the highest average engagement frequency, reading 43.2% of the emails.

In contrast, no statistically significant difference was observed in the engagement frequency between employees with good and poor physical activity habits (see Table 2; *p* = 0.807). 

We also analyzed engagement frequency by job type, differentiating between sedentary work, standing work, and physical work. On average, the highest engagement frequency was observed in employees performing mostly sedentary work (42.6%; see Table 2). This trend is presented in Figure 3. The engagement frequency among the other two groups of employees was lower by 13 percentage points (see Table 2).

We confirm statistically significant differences between the different types of work (*p* < 0.001; see Table 2). A further multiple comparison revealed statistically significant differences between sedentary and standing work (*p* < 0.001) and sedentary and physical work (*p* < 0.001). 

A comparison of engagement frequencies by type of work and by gender provides an interesting insight. As shown in Figure 4, men engaged in either standing or physical work exhibited a fairly similar right-skewed distribution of engagement frequency, while for women engaged in standing work, the distribution was more right-skewed. These women also had the lowest average engagement frequency (27.4%). The highest engagement frequency was observed among both women and men who performed sedentary work, with men having a significantly higher average engagement frequency (46.8% compared to 37.9%). For both men and women, we confirmed that the differences in engagement frequency across different types of work were statistically significant (at *p* = 0.051 for women and *p* = 0.001 for men). 

We further compared the engagement frequency by type of work and dietary habits. As shown in Figure 5, the distribution of engagement frequencies is skewed to the right for all types of work but is more pronounced among employees engaged in standing and physical work. In both groups with predominantly poor and good dietary habits, the average engagement frequency was highest among employees engaging in sedentary work (40% and 44.3%, respectively). Moreover, we find statistically significant differences in engagement frequencies between the different types of work for both employees with poor dietary habits (*p* = 0.000) and those with good dietary habits (*p* = 0.014). 

Finally, we sought to determine whether there was a relationship between participants’ engagement frequency and their BMI. Our results showed no statistically significant association between the two. However, we found a statistically significantly higher association between engagement frequency and BMI in men (the average BMI was 25.6 for women and 26.8 for men; *p* = 0.000), older workers (27.4 vs. 25.3 for middle-aged and 24.3 for young employees; *p* = 0.003), and employees performing standing work (27.2 vs. 26.6 for physical work and 25.5 for sedentary work; *p* = 0.065). Figure 6 provides a more in-depth analysis of the BMI distribution across the types of work and gender. The distribution of BMI is particularly right-skewed for men performing sedentary work, with a spike in the area of the normal weight range. Our analysis confirmed that the men who performed sedentary work had a statistically significantly lower BMI (on average, 24.8) than those engaged in the other two types of work (26.5 for physical work and 26.9 for standing work, on average) at *p* < 10%. While the BMI distribution for women did not include such high spikes, we found that women performing standing and physical work had BMI distributions more skewed to the left. For women, there were no statistically significant differences in BMI between the different types of work. 

Finally, we used a multiple linear regression to examine the association between engagement frequency and dietary and physical activity habits, types of work, and demographic characteristics (Table 3). 

The results indicate that, when controlling for demographic characteristics and job type, employees with good dietary habits exhibited a statistically significant higher engagement frequency compared to employees who reported poor dietary habits, with their engagement frequency being higher by 6.02 percentage points (*p* < 0.1). There was also a statistically significant relationship between engagement frequency and type of work. Specifically, the engagement frequency of employees who performed standing or physical work was lower by 13.22 or 11.33 percentage points, respectively, than the engagement frequency of workers who performed sedentary work (*p* < 0.001). The regression coefficient was also statistically significant (*p* < 0.05) for men. For the other independent variables, the regression coefficients were not statistically significant. 

## 4. Discussion

Our study examined the behavioral patterns of 370 employees from 26 companies, of whom 57.3% were employed in sedentary work, 19.5% in standing work, and 23.2% in physical work. Contrary to many of the findings in the literature which identify sedentary occupations as the greatest risk for obesity [26,27] and unhealthy dietary behavior [28], we found that sedentary workers in our study had a lower BMI index than workers in more physically active occupations. The BMI distribution across different occupations and genders (Figure 6) shows a clear right skew for men in sedentary occupations, with a peak in the normal weight range. The men engaged in sedentary work had a statistically significantly lower BMI (an average of 24.8) compared to their counterparts engaged in physical work (an average of 26.5) and standing work (an average of 26.9). We also controlled the differences in BMI for age but we did not find any significant differences. While the BMI distribution for women does not show such high peaks, we can observe a pronounced left skew of the BMI distribution for women engaged in standing and physical work. No statistically significant differences were observed in the women’s BMI across different occupational groups, which confirms the results of previous studies [27].

To assess participants’ engagement frequency, we used e-platform monitoring to determine the proportion of employees participating in the web-based campaign who actually read the materials sent. The percentage of e-mails read (engagement frequency) was higher among men (Table 2). We wondered whether the men might have read the emails more often because they are more active online. But women seemed to be more engaged in using internet services about health and education [28]. However, some research shows that employees of both genders find the internet equally useful, except that woman develop more anxiety when using the internet [29]. Perhaps this explains their lower engagement in reading, browsing various links, and watching videos in our web-based campaigns. In terms of age demographics, older employees demonstrated a significantly higher average engagement frequency than their younger counterparts, with a mean difference exceeding 10 percentage points. This trend was also observed in the distribution of engagement frequency across age groups (Figure 1). This result was particularly surprising given the higher prevalence of internet use among the younger population [30,31]. 

In terms of dietary habits, the results show that the engagement frequency of employees with poor dietary habits was lower than that of employees with good dietary habits, with a statistically significant difference between the means. Further stratification by gender (see Figure 3) shows that the engagement frequency for both women and men with poor dietary habits was lower (28.9% and 35.6%, respectively, on average) than that of employees with good dietary habits. Interestingly, employees with the highest average engagement frequency were men with good eating habits (43.2% of emails read, on average). In contrast, no statistically significant difference was observed between the engagement frequency of employees with good and poor physical activity habits (see Table 2). It is possible that the results are influenced by the participants’ level of education and personal income, which were not controlled for. It has been proven that education and personal income affect one’s eating habits. According to a Slovenian study, men with a low level of education and low income have the worst diet. They skip breakfast, eat too much red meat, and consume too much saturated fat [32]. But we must take into account that in the companies in question, mainly due to their small size, the structure of employees is very heterogeneous in terms of the tasks they perform, but not so much in terms of education. The vast majority of employees in the sample have secondary or higher education in all groups, including those working sedentary, standing, and physically demanding jobs. It is well known that workers in more physically active occupations with lower levels of education and income (truck drivers, construction workers, etc.) are “hard to reach” through digital health interventions, especially males, which was also found in our research [33,34]. This reduces the possibility of these differences.

When examining engagement frequency by type of work, distinguishing between sedentary, standing, and physical work, we found that, on average, the engagement frequency was highest among employees engaged in predominantly sedentary work (42.6%; see Table 2). It is worth contemplating the notion that individuals in sedentary roles, particularly those working at a computer, might have enhanced accessibility to read emails throughout their workday, distinguishing them from individuals engaged in more physically demanding tasks. This consideration could shed light on potential variations in email engagement based on the nature of one’s job and responsibilities. 

A comparison of the engagement frequency by type of work and gender offers an interesting insight. While men performing either standing or physical work recorded a fairly similar right-skewed distribution of their engagement frequency, this distribution was more right-skewed for women performing standing work (Figure 4). Women engaged in standing work also recorded the lowest average engagement frequency (27.4%). Engagement frequency was highest for both women and men performing sedentary work, although men had a much higher average engagement frequency (46.8% compared to 37.9%). 

After controlling for demographic characteristics and types of work, employees with good dietary habits were shown to exhibit a statistically significant higher engagement frequency than employees who reported poor dietary habits. On average, their engagement frequency was 6.02 percentage points higher. Employees with better habits appear to read more and exhibit higher levels of awareness commitment to different tasks. This aligns with the findings by Virtanen et al., who reported that the more frequent consumption of healthy foods is associated with greater work engagement, regardless of the consumption of unhealthy foods [35]. There was also a statistically significant relationship between the engagement frequency and type of work. More specifically, the engagement frequencies of workers engaged in standing or physical work were 13.22 and 11.33 percentage points lower, respectively, than that of workers engaged in sedentary work. The regression coefficient was also statistically significant for men. For the other independent variables, the regression coefficients were not statistically significant.

Our findings suggest that older male workers with good dietary habits and sedentary work had the highest engagement frequency in web-based campaigns. These workers also had a reasonable BMI. This suggests that the web-based campaign method seems less suitable for workers with poorer dietary habits and predominantly standing and physically demanding work. On the other hand, unfortunately, we did not identify the study population according to education and personal income. Nevertheless, current interventions promoting a healthy diet and physical activity and policies seem to fail in reaching those in most need of health-enhancing physical activity and a healthy diet: individuals of lower socioeconomic status [36]. However, in our study, the participants with sedentary occupations come from well-known, successful companies, are usually more selective regarding their education (mechanical engineers, computer engineers, etc.), and have higher levels of income than the participants with more physically active occupations (truck drivers, construction workers, etc.). It is well known that the latter group of workers is “hard to reach” through digital health interventions, especially males [33,34]. These findings are important for the development of workplace health promotion policies and practices.

This study acknowledges several limitations that warrant consideration. The utilization of a cross-sectional, correlational design constrains our ability to make definitive assertions about cause–effect relationships. Our research design may not have accounted for all relevant factors, potentially introducing uncontrolled variables. Consequently, the correlations observed should not be interpreted as implying causation, as confounding factors (such as education) may influence the observed relationships. Furthermore, despite the study spanning a 2-year period, there is a possibility that dynamic processes were overlooked. It is imperative to recognize the presence of selection bias, which limits the generalizability of our findings to broader populations. The study employed a non-probabilistic availability sample comprising 370 workers from companies that voluntarily participated in the WHPP. As a result, our research findings can be generalized only to similar small- and medium-sized enterprises (SMEs) in Slovenia that exhibit a predisposition towards engaging in WHPPs. Finally, the study acknowledges potential measurement issues that could impact the validity of the observed correlations. Specifically, the operationalization of ‘emails read’ poses a potential challenge. While we defined an email as ‘read’ only when the email was opened and the content was rated, it is important to note that the participants’ reading comprehension and consideration of the content cannot be definitively determined.

Lastly, many employers were reluctant to provide data; all data were self-reported and the study was conducted during the lockdown due to COVID-19. All these factors could have potentially distorted our final results. 

## 5. Conclusions

In Slovenia, as in the rest of the world, we are confronted with the problem of obesity and obesity-related diseases. The workplace represents a key focus for public health initiatives aimed at combating obesity in the working population. Our study delved into the behavioral patterns of 370 employees across 26 SMEs, shedding light on their engagement frequencies in web-based educational campaigns, dietary and physical activity habits, and the correlation of these with demographic factors. Several noteworthy findings emerged, challenging some established notions in the literature. Men with sedentary work demonstrated a higher engagement in web-based campaigns, with older employees participating more actively than expected. Healthy dietary habits correlated with increased engagement, emphasizing lifestyle influences. Despite some limitations, our findings contribute valuable insights for workplace health promotion policies and practices. Our results underscore the importance of tailoring interventions to specific occupational groups, considering demographic factors, and recognizing the multifaceted influences on employee engagement in health-related initiatives.

## Figures and Tables

**Figure 1 nutrients-15-05129-f001:**
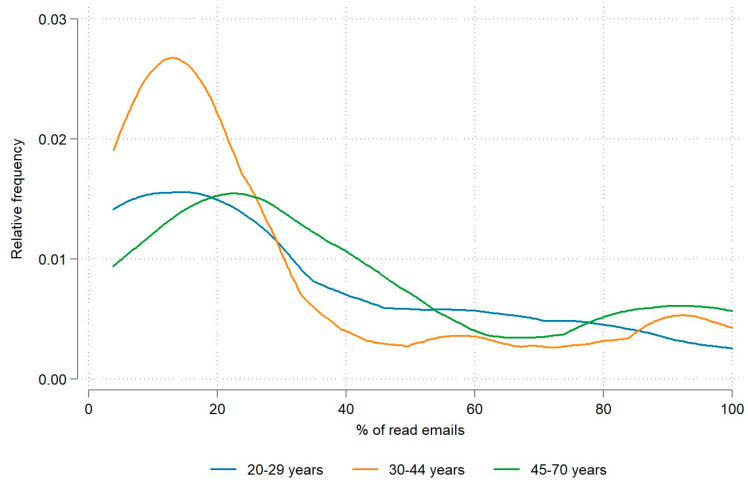
Relative engagement frequency by age groups.

**Figure 2 nutrients-15-05129-f002:**
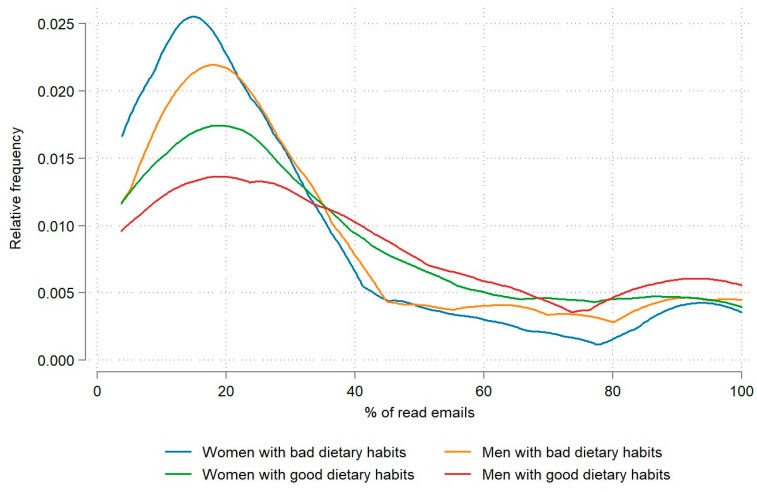
Relative engagement frequency by dietary habits and gender.

**Figure 3 nutrients-15-05129-f003:**
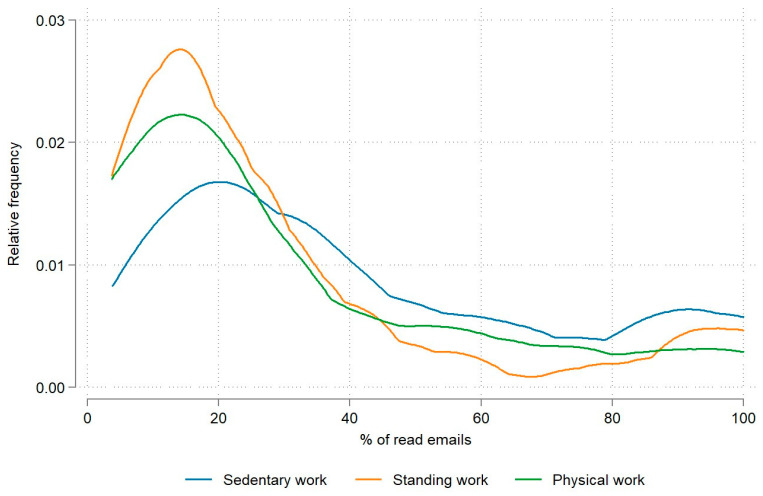
Relative engagement frequency by type of work.

**Figure 4 nutrients-15-05129-f004:**
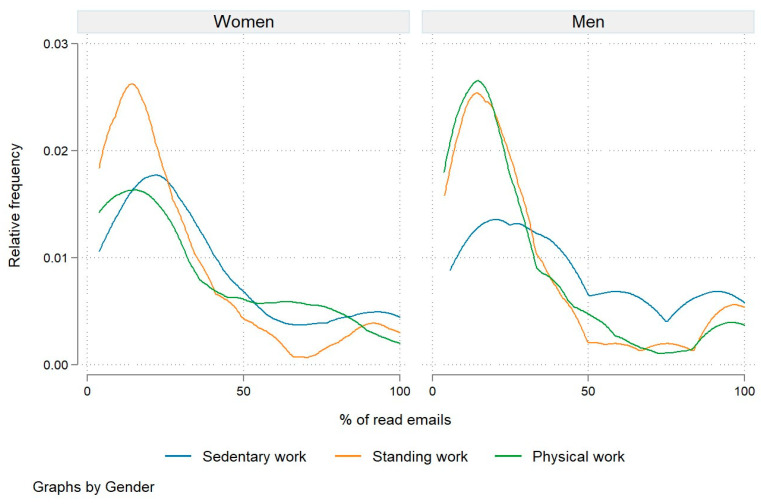
Relative engagement frequency by type of work and by gender.

**Figure 5 nutrients-15-05129-f005:**
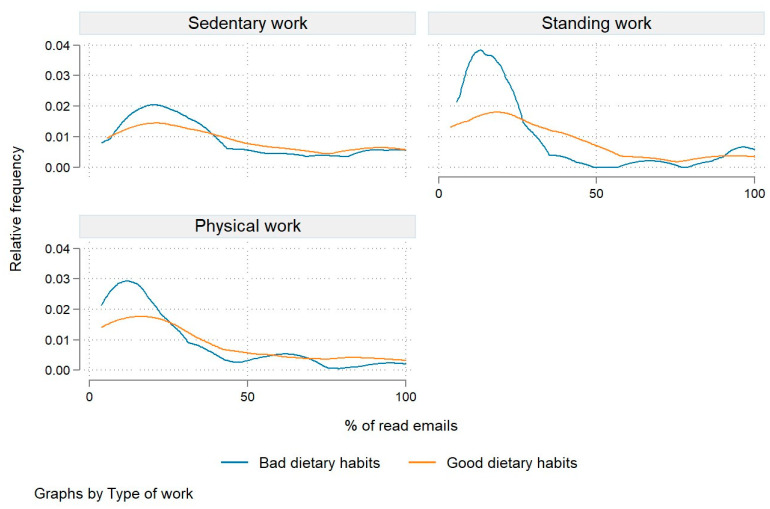
Relative engagement frequency by type of work and dietary habits.

**Figure 6 nutrients-15-05129-f006:**
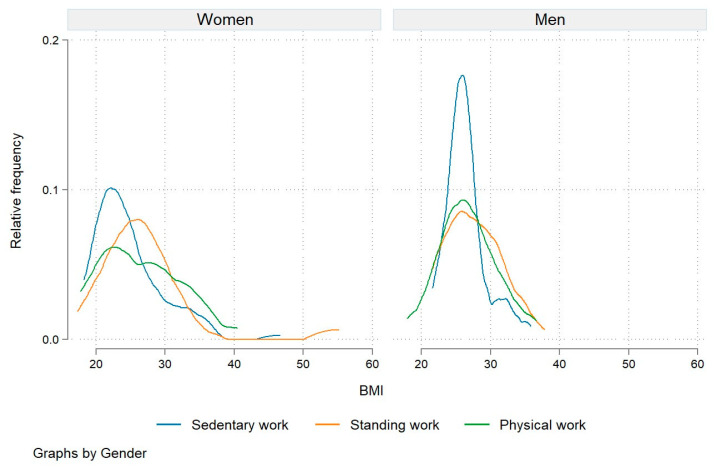
Relative frequency of BMI by job type and gender.

**Table 1 nutrients-15-05129-t001:** Employee characteristics by gender.

Characteristics	Female n (%)	Male n (%)	Total n (%)
Employees	172 (46.5)	198 (53.5)	370 (100)
Sociodemographic characteristics Age groups (years)			
20–29	16 (9.3)	22 (11.1)	38 (10.3)
30–44	49 (28.5)	43 (21.7)	92 (24.8)
45–70	55 (32.0)	40 (20.2)	95 (25.7)
Data not available	52 (30.2)	93 (50.0)	145 (39.2)
Type of work			
Sedentary	99 (57.6)	113 (57.1)	212 (57.3)
Standing	38 (22.1)	34 (17.2)	72 (19.5)
Physical	35 (20.4)	51 (25.7)	86 (23.2)
BMI (kg/m^2^) ^1^			
Underweight (<18.5)	3 (1.7)	1 (0.5)	4 (1.1)
Normal weight (18.5–24.99)	66 (38.4)	33 (16.7)	99 (26.8)
Pre-obesity (25–29.99)	25 (16.3)	56 (28.3)	84 (22.7)
Obesity—class I (30–34.99)	19 (11.0)	17 (8.6)	36 (9.7)
Obesity—class II (35–39.99)	2 (1.2)	3 (1.5)	5 (1.3)
Obesity—class III (≥40)	3 (1.7)	0	3 (0.8)
Data not available	51 (29.7)	88 (44.4)	139 (37.6)
Unhealthy dietary habits			
Eating ≤ 2 meals per day	42 (24.4)	79 (39.9)	121 (32.7)
Eating < 1 vegetable a day	47 (27.5)	74 (37.4)	121 (32.8)
Eating < 1 fruit a day	55 (32.0)	94 (47.5)	149 (40.3)
Eating ≥ 3 servings of red meat/week	55 (32.2)	111 (56.1)	166 (45.0)
Eating ≤ 3 fish/month	136 (79.1)	124 (62.6)	260 (70.3)
Eating ≤ 3 servings of whole grain/month	84 (48.8)	114 (57.6)	198 (53.5)
Eating ≥ 1 serving of fried food/week	33 (19.2)	82 (41.6)	115 (31.2)
Drinking ≥ 4 sugary beverages/week	21 (12.2)	37 (18.8)	58 (15.7)
Predominantly good dietary habits ^1^	116 (67.4)	99 (50.0)	215 (58.1)
Unhealthy PA habits			
Vigorous intensity PA < 75 min/week	43 (42.6)	64 (44.4)	107 (43.7)
Moderate intensity PA < 150 min/week	88 (60.3)	102 (61.5)	190 (60.9)
Predominantly good PA habits ^2^	92 (59.7)	106 (58.9)	198 (59.3)

Legend: BMI, body mass index; n, the number of subjects; %, percentage of subjects; PA, physical activity. Notes: ^1^ The variable Predominantly good dietary habits is derived from variables on unhealthy dietary habits. An individual’s dietary habits were identified as predominantly good if the individual reported five or more healthy dietary habits. ^2^ An individual is classified as having predominantly good PA habits if they engage in at least 75 min of vigorous exercise per week or at least 150 min of moderate exercise per week.

**Table 2 nutrients-15-05129-t002:** Descriptive statistics and inferential statistics related to engagement frequency categorized by demographic groups, types of work, dietary habits, and physical activity levels.

Group	Descriptive Statistics		Inferential Statistics	
	n	Mean ± SD	Wilcoxon–Mann–Whitney Test	
			z-Value	*p*
Gender				
Female	172	34.3 ± 28.7	–1.613	0.107
Male	198	39.4 ± 31.6		
Dietary habits				
Poor dietary habits	155	33.2 ± 29.1	–2.078	0.038 **
Good dietary habits	215	39.8 ± 31.1		
Physical activity				
Poor PA habits	136	36.5 ± 30.6	–0.424	0.672
Good PA habits	198	37.3 ± 29.8		
	n	Mean ± SD	Kruskal–Wallis H test	
			χ^2^(2)	*p*
Age group (years)				
20–29	38	31.5 ± 30.2	10.992	0.041 **
30–44	92	30.5 ± 30.1		
45–70	95	41.9 ± 32.3		
Type of work				
Sedentary	212	42.6 ± 31.2	22.832	0.000 ***
Standing	72	29.5 ± 28.0		
Physical	86	29.4 ± 27.4		

Legend: PA, physical activity. Statistical significance: ** *p* < 0.05; *** *p* < 0.001.

**Table 3 nutrients-15-05129-t003:** Results of the linear regression.

	Dependent Variable: Engagement Frequency
Gender (baseline: women)	
Men	7.537 **
	(3.391)
Age (baseline: 20–29 years)	
30–44 years	–2.509
	(6.145)
45–70 years	7.465
	(6.193)
Data not available	4.265
	(5.947)
Dietary habits (baseline: poor dietary habits)	
Good dietary habits	6.016 *
	(3.460)
PA habits (baseline: poor PA habits)	
Good PA habits	–1.651
	(3.434)
Job type (baseline: sedentary work)	
Standing work	–13.215 ***
	(3.989)
Physical work	–11.332 ***
	(4.206)
Constant	32.211 ***
	(6.381)
Number of observations	334
R2	0.08

Legend: PA, physical activity. Note: Robust standard errors in parentheses. Statistical significance: * *p* < 0.1; ** *p* < 0.05; *** *p* < 0.001.

## Data Availability

The results will be communicated to the participants and other relevant groups via publications (open access will be prioritized) and presentations, including webinars.

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
