# Peer review of "Factors Influencing the Degree of Employee Involvement in Preventive Nutrition and Physical Activity Web-Based Programs in Medium and Small Enterprises"

_nutrients, 2023, doi:10.3390/nu15245129_

Round 1

Reviewer 1 Report

Comments and Suggestions for Authors

The present study examined the effect of a web-based educational intervention campaign to promote healthy eating habits and physical activity to reduce sick leave absences in small-to medium size enterprises. The study analyzes the association between engagement frequency with the campaign and work type and multiple demographic aspects. It gives a comprehensive overview and is well structured. However, I have the following comments and concerns:

Title: Why does the title focus on healthy diets? If I understood it correctly the education program was on diet as well as physical activity. Also it was not about promoting healthy diet in the workplace, i.e. providing healthy lunch options at the canteen etc. Reading the title I expected a different focus/content of the study. It would be helpful if you could align the title with the focus of the paper.

Abstract: Please add the duration of the study period as a central information.

Introduction:

I'm missing some justification for the separation into the three work groups (sedentary, standing, physical). Please elaborate more on the reasoning behind expecting differential differences in these groups.

Were participants actually sent emails to their personal email addresses, or were they sent to their work email addresses, accessible and readable during working hours and at the workplace? I would assume that this makes a big difference. Please specify.

Method:

I have doubts about the operationalization of "Email opened = Email read." Just marking an email as opened doesn't necessarily mean its content was read. Was this check in another way?

It is impressive that the study analyzes data over a period of 5 years. However, measuring the impact on NCDs within this timeframe may be challenging. NCDs arise from long-term unfavorable life and lifestyle conditions. Five years are relatively short in this context, raising the question of whether meaningful associations can be measured?

In the methods and research questions, it appears that the correlation with sick leave and absence days is an important central question of the study. I indeed find this to be a crucial aspect, as it represents an objective indicator. However, in the results, it is only presented very briefly and lacks differentiation.

Importantly, were baseline differences between participating and non-participating companies controlled for? Otherwise, drawing conclusions might be difficult.

Methods/Results:

Please clearly separate methods and results. Explain the statistical approach in the methods, allowing the results section to focus solely on the central findings.

Clarify why "own calculation" is noted as the source for figures; are there figures based on other calculations?

Results:

The results about "significant differences between different types of work" and in relation to the differences in work type and dietary habits may indicate that these individuals sit at the computer and read emails not because they are more interested but because it fits better into their work routine. If I read and write emails all day long it is much easier to read another email than when my normal work day does not include sitting down at a computer reading emails anyway. Did you control for it or at least ask about how it? I find this to be a crucial point for discussion.

The result stating "men who performed sedentary work had a lower BMI" needs clarification regarding age. Was age controlled for? Considering that BMI is lower in younger ages, what is the age distribution in the three workgroups? Please discuss this in the discussion.

Generally, it think it is challenging to examine individual aspects (work type, dietary habit, PA, etc.) separately without controlling for age, BMI, etc. Without knowledge of the influence of these variables, meaningful statements or conclusions are difficult to draw. Or at least it needs to be discussed and mentioned in the discussion.

Results on sick leave absence: Where are Figure 7 and Figure 8? These graphics were not included in my manuscript.

Further, a more detailed analysis of various causes and diseases would have been interesting in relation to sick leave and absence days. Those results are very brief.

The paragraph about sick leave absences includes first explanations. Please keep explanatory and discussion approaches to be part of the discussion, not the results.

Discussion:

In the discussion, it is stated, "We developed web-based content on health eating and PA." ? Reading the methods, I understood that only an existing platform was used. Please specify.

Are the 370 participants from the results the participants from the 26 companies that also participated in nutrition and PA? This remains unclear - please elaborate.

Overall, the discussion reads still much of a mere repetition of the results. The results, including used statistical tests, are repeated in detail. So far, I would say that the discussion hasn't made much of a significant independent contribution to the paper, except for a recapitulation of the results. Please provide a more detailed and elaborated integration of the results into existing research, including a critical examination and the presentation of implications.

It would be helpful to have an initial short section with the main results of the study and then subsequently more actual discussion and critical examination of the results.

I miss in the discussion the question of whether the results might also be explained by the fact that individuals engaged in sedentary work can more easily read emails during their workday.

Further, to what extent are those differences possibly explained by educational level? Are individuals with sedentary jobs more educated than those with physical work? Was this controlled for? Please specify.

In the discussion, the difference in engagement frequency between women and men is explained by the fact that men use the internet more than women. However, is reading emails sent directly to an individual's account really considered internet usage? The source you cite to backup your argument discusses internet usage among high school students aged 12 to 18 and the link between internet usage and school activities. This does not correspond neither to the age group nor activity in the presented study. Therefore, the source and the drawn explanation are not appropriate for the results of the presented study. Similarly, I also question the second source regarding differences in internet usage. Firstly, it is a very old study, and the source itself concludes that "the gender gap in being online disappeared by 2000." Secondly, there are newer studies by the same author.

In lines 449-451, implications are briefly addressed. Discussion potential implications of the results would be helpful for future research and intervention implementation. What are explanatory approaches for the found results and what does this mean for future interventions, etc.? Please elaborate more.

Limitations are very short. Please elaborate more. I would say that especially the operationalization of emails read, the potential cofounding effect of education and the question of accessibility of emails during work are at least worth discussion as potential limitations of the study results.

Author Response

Our responses to each point raised by the Reviewer 1 are marked with arrows below the comments (à). Changes in the manuscript are marked in red.

We trust that we have adequately answered all the comments and made appropriate changes. We are grateful for the constructive recommendations that resulted in an improved text and a better understanding of the revised manuscript.

In conclusion, we hope that the revised manuscript is now acceptable for publication in your respected journal. Thank you for your attention and interest.

Reviewers' comments 1:

  1. Title: Why does the title focus on healthy diets? If I understood it correctly the education program was on diet as well as physical activity. Also it was not about promoting healthy diet in the workplace, i.e. providing healthy lunch options at the canteen etc. Reading the title I expected a different focus/content of the study. It would be helpful if you could align the title with the focus of the paper.

à As requested by the Reviewer 1, the present manuscript was reviewed and the title was corrected. The new title is: Engagement frequency in web-based nutrition and physical activity educational campaign.

  1. Abstract: Please add the duration of the study period as a central information.

à As requested by the Reviewer 1, the manuscript now contains the duration of the study. We add the sentence: The project lasted from 2020 to 2022. 

  1. Introduction: I'm missing some justification for the separation into the three work groups (sedentary, standing, physical). Please elaborate more on the reasoning behind expecting differential differences in these groups.

à In order to categorize three work groups survey respondents, we utilized the FAO/WHO/UNU [24] classification and Ministry of Health of the Republic of Slovenia Guidelines for Healthy Eating for Employees [25], which is predicated on the total energy expenditure and physical activity level of adults. This classification system, comprising Sedentary or light activity lifestyle, Active or moderately active lifestyle, and Vigorous or vigorously active lifestyle, was adapted to the context of the working environment. The decision to employ the FAO/WHO/UNU (2004) classification stems from its relevance in assessing different levels of physical activity among adults. Specifically, it allows for the identification of Sedentary or light activity, Active or moderately active, and Vigorous or vigorously active individuals based on their total energy expenditure. This adaptation to the working environment is grounded in the hypothesis that distinct types of work are associated with varying energy expenditures, which, in turn, correlate with diverse dietary needs and habits.

  1. Human Energy Requirements: Report of a Joint FAO-WHO-UNU Expert Consultation : Rome, 17-24 October 2001; FAO food and nutrition technical report series; United Nations University: Rome, 2004; ISBN 978-92-5-105212-9.
  2. Ministry of health, Republic Slovenia. Healthy eating guidelines for workers in the workplace. Available online in slovenian language: https://www.gov.si/assets/ministrstva/MZ/DOKUMENTI/Preventiva-in-skrb-za-zdravje/Varovanje-in-krepitev-zdravja/zdravje-na-del-mestu/Smernice_zdravega_prehranjevanja_delevcev.pdf (accessed on 26 Nov 2023).

  1. Were participants actually sent emails to their personal email addresses, or were they sent to their work email addresses, accessible and readable during working hours and at the workplace? I would assume that this makes a big difference. Please specify.

à Thank you for highlighting the importance of distinguishing between different email accounts. Throughout the project/research, we explicitly requested respondents to provide email addresses that they access daily, preferably through their mobile phones. As a result, the majority of participants provided us with their personal email accounts. In response to Reviewer 1's suggestion, we enhanced information clarifying that the web-based campaign was sent to personal email addresses. It's worth noting that on page 3 of the methods section, the following sentence was already included: 'Each employee received one unit of content per week to their personal email.”

  1. Method: I have doubts about the operationalization of "Email opened = Email read." Just marking an email as opened doesn't necessarily mean its content was read. Was this check in another way?

àThank you for highlighting this highly relevant concern. In response to Reviewer 1's request, we have augmented the manuscript to include detailed information on the operationalization of 'Email opened = Email read.' Each email sent during the study contained a link to content on a landing page, which, in turn, included further links to content or video. At the conclusion of each landing page, participants were prompted to provide a content rating. We established that an email was 'read' only for those individuals who actively gave a content rating. Importantly, it is acknowledged that this operationalization does not confirm whether the email was read and considered in full. We delve into this aspect further in the limitations section of the study.

  1. It is impressive that the study analyzes data over a period of 5 years. However, measuring the impact on NCDs within this timeframe may be challenging. NCDs arise from long-term unfavorable life and lifestyle conditions. Five years are relatively short in this context, raising the question of whether meaningful associations can be measured?

à As advised by Reviewer 1, we changed the text and removed the sentence that suggests a connection. We also add the sentence: The entire project started on 1 September 2017 and was completed on 30 September 2022, and the campaign in the field of nutrition and physical activity lasted from 2020 to 2022. 

  1. In the methods and research questions, it appears that the correlation with sick leave and absence days is an important central question of the study. I indeed find this to be a crucial aspect, as it represents an objective indicator. However, in the results, it is only presented very briefly and lacks differentiation.

à The utilization of a descriptive, cross-sectional, correlational design constrains our ability to make definitive assertions about cause–effect relationships. Our research design may not have accounted for all relevant factors, potentially introducing uncontrolled variables. Consequently, the correlations observed should not be interpreted as implying causation, as confounding factors (such as education) may influence the observed relationships. Furthermore, despite the study spanning a 2-year period, there is a possibility that dynamic processes were overlooked.

  1. Importantly, were baseline differences between participating and non-participating companies controlled for? Otherwise, drawing conclusions might be difficult.

àTo address the reviewer's observation regarding the need to control for differences between participating and non-participating companies, it's important to note that our research design did not include a control group. We have taken steps to clarify this aspect by extensively revising the Research Limitations section, explicitly stating that the design did not envisage the inclusion of a control group. This information is now articulated in a more detailed manner to provide a clear understanding of the study's limitations. We appreciate the reviewer's attention to this matter and believe that the revised section adequately addresses the raised concern.

  1. Methods/Results: Please clearly separate methods and results. Explain the statistical approach in the methods, allowing the results section to focus solely on the central findings.

à As advised by Reviewer 1, we separate methods and results.

  1. Clarify why "own calculation" is noted as the source for figures; are there figures based on other calculations?

à As advised by Reviewer 1, we deleted “own calculations” from figures.

  1. Results: The results about "significant differences between different types of work" and in relation to the differences in work type and dietary habits may indicate that these individuals sit at the computer and read emails not because they are more interested but because it fits better into their work routine. If I read and write emails all day long it is much easier to read another email than when my normal work day does not include sitting down at a computer reading emails anyway. Did you control for it or at least ask about how it? I find this to be a crucial point for discussion.

àThank you for your observation, which is certainly valid. We responded to this observation in depth in the CONCLUSION section, while addressing the future research focus.

  1. The result stating "men who performed sedentary work had a lower BMI" needs clarification regarding age. Was age controlled for? Considering that BMI is lower in younger ages, what is the age distribution in the three workgroups? Please discuss this in the discussion.

à Thank you for the suggestion. The age was controlled for. However, we did not find any significant differences. In the DISCUSSION section we have added some thoughts / interpretations addressing this dilemma.

  1. Generally, it think it is challenging to examine individual aspects (work type, dietary habit, PA, etc.) separately without controlling for age, BMI, etc. Without knowledge of the influence of these variables, meaningful statements or conclusions are difficult to draw. Or at least it needs to be discussed and mentioned in the DISCUSSION.

à Please see previous response.

  1. Results on sick leave absence: Where are Figure 7 and Figure 8? These graphics were not included in my manuscript.

à As advised by Reviewer 1, we added Figure 7 and Figure 8. We sincerely apologize for the oversight regarding the absence of two figures in the uploaded manuscript on the journal's website. Unfortunately, we currently do not have a clear explanation for this discrepancy, as the version on our local PCs includes the mentioned figures. We understand the importance of a comprehensive submission and deeply regret any inconvenience this may have caused.

  1. Further, a more detailed analysis of various causes and diseases would have been interesting in relation to sick leave and absence days. Those results are very brief.

à Thank you for your suggestion, but we cannot address this in the article. However, it is an important topic for future research.

  1. The paragraph about sick leave absences includes first explanations. Please keep explanatory and discussion approaches to be part of the discussion, not the results.

à Indeed, your observation is to the point. This part / text was moved to the DISCUSSION section.

Discussion:

  1. In the discussion, it is stated, "We developed web-based content on health eating and PA." ? Reading the methods, I understood that only an existing platform was used. Please specify.

àAs requested by the Reviewer 1, the Discussion section of the revised manuscript now contains additional specifications, where we rewrote the text, more specifically conveying that both platform and the content were newly developed.

  1. Are the 370 participants from the results the participants from the 26 companies that also participated in nutrition and PA? This remains unclear - please elaborate.

à As requested by the Reviewer 1, the Discussion section of the revised manuscript now contains additional description: Our study examined the behavioral patterns of 370 employees, from 26 companies, ...

  1. Overall, the discussion reads still much of a mere repetition of the results. The results, including used statistical tests, are repeated in detail. So far, I would say that the discussion hasn't made much of a significant independent contribution to the paper, except for a recapitulation of the results. Please provide a more detailed and elaborated integration of the results into existing research, including a critical examination and the presentation of implications.

à Correct observation. Thank you for pointing out this issue. The points you have advised were incorporated into rewritten DISCUSSION section.

  1. It would be helpful to have an initial short section with the main results of the study and then subsequently more actual discussion and critical examination of the results.

à We agree. As suggested, we have condensed the section discussing the results and expanded the scope of the critical discussion.

  1. I miss in the discussion the question of whether the results might also be explained by the fact that individuals engaged in sedentary work can more easily read emails during their workday.

à Thank you for raising a valid and relevant point. To enhance the clarity of our reasoning and the foundation of our conclusion, we have explicitly stated in the manuscript that, throughout the project/research, we asked respondents to provide email addresses that they access daily, preferably through their mobile phones. Consequently, the majority of participants provided us with their personal email accounts. This ensures that the content was available to the vast majority of participants both during and outside of work hours, independent of Internet access availability during work hours. Of course, there is still the possibility of the mentioned influence, which we wrote in the limitations.

  1. Further, to what extent are those differences possibly explained by educational level? Are individuals with sedentary jobs more educated than those with physical work? Was this controlled for? Please specify.

à It is possible, because it has been proven that education affects the eating habits. But we must take into account that in companies in question, mainly due to their small size, the structure of employees is very heterogeneous in terms of the tasks they perform, but not so much in terms of education. The vast majority of employees in the sample have secondary and higher education, which reduces the possibility of these differences. We included this reasoning in the discussion.

  1. In the discussion, the difference in engagement frequency between women and men is explained by the fact that men use the internet more than women. However, is reading emails sent directly to an individual's account really considered internet usage? The source you cite to backup your argument discusses internet usage among high school students aged 12 to 18 and the link between internet usage and school activities. This does not correspond neither to the age group nor activity in the presented study. Therefore, the source and the drawn explanation are not appropriate for the results of the presented study. Similarly, I also question the second source regarding differences in internet usage. Firstly, it is a very old study, and the source itself concludes that "the gender gap in being online disappeared by 2000." Secondly, there are newer studies by the same author.

à As requested by the Reviewer 1, we have replaced the sentences and the more recent study. The text reads as follows: We wondered whether men might read email more often because they are more active online? But women seem to be more engaged in using Internet services about health and education [29]. However, some research shows that employees of both genders find the internet equally useful, except that woman develop more anxiety when using the internet [30]. Perhaps this explains the lower engagement in reading, browsing various links and watching videos of our web-based campaigns.

  1. Garín-Muñoz, T.; Pérez-Amaral, T.; Valarezo, Á. Evolution of the internet gender gaps in Spain and effects of the Covid-19 pandemic. Telecomm Policy 2022, 46, 102371. doi: 10.1016/j.telpol.2022.102371
  2. Zhang, Y. Age, gender, and Internet attitudes among employees in the business world. Computers in Human Behavior 2005, 21, 1–10. doi: 10.1016/j.chb.2004.02.006.

  1. In lines 449-451, implications are briefly addressed. Discussion potential implications of the results would be helpful for future research and intervention implementation. What are explanatory approaches for the found results and what does this mean for future interventions, etc.? Please elaborate more.

àWe have expanded the proposed measures and research arising from our results. The added text reads: On the other hand, unfortunately, we did not identify study population according to education and personal income. Nevertheless, current physical activity and healthy diet promoting interventions and policies seem to fail in reaching those in most need of health-enhancing physical activity and choosing healthy diet: individuals of lower socioeconomic status (33). However, in our study those participants with sedentary occupation, come from well-known, successful companies and are usually more selective regarding their education (mechanical engineers, computer engineers, etc.) and have higher income than those participants in more physically active occupations (truck drivers, construction workers, etc.). It is well known that the latter group of workers is “hard-to-reach” with digital health interventions, especially male (34, 35).

  1. Ono, H.; Zavodny, M. Gender and the Internet. Social Science Quarterly 2003, 84, 111–121. doi: 10.1111/1540-6237.t01-1-8401007. Rasmussen, C. L.; Dumuid, D.; Hron, K. Day-to-day pattern of work and leisure time physical behaviours: are low socioeconomic status adults couch potatoes or work warriors?. BMC Public Health. 2021, 21, 1342. doi:10.1186/s12889-021-11409-0
  2. Sendall, M. C.; McCosker, L. K.; Crane, P.; Rowland, B.; Fleming, M.; Biggs, H. C. Using Facebook for Health Promotion in "Hard-to-Reach" Truck Drivers: Qualitative Analysis. J Med Internet Res. 2018, 20, e286. doi:10.2196/jmir.9689
  3. Seaton, C. L.; Bottorff, J. L.; Caperchione, C. M.; Johnson S. T.; Oliffe, J. L. The Association Between Men's Heath Behaviors and Interest in Workplace Health Promotion. Workplace Health Saf. 2020, 68, 226-235. doi:10.1177/2165079919885957

  1. Limitations are very short. Please elaborate more. I would say that especially the operationalization of emails read, the potential cofounding effect of education and the question of accessibility of emails during work are at least worth discussion as potential limitations of the study results.

à To address the valuable observations from both reviewers, we have extensively revised the study’s limitations section, presenting it as a distinct section. There, we thoroughly addressed key aspects including (a) limitations related to the research design, (b) considerations regarding selection bias, and (c) a detailed exploration of measurement limitations. This restructuring aims to provide a more in-depth and transparent discussion of the study's constraints. We appreciate the reviewers' insights, and we believe this enhanced presentation strengthens the overall quality of the manuscript.

Reviewer 2 Report

Comments and Suggestions for Authors

The search for ways to increase awareness and awareness of patients about the prevention of chronic non-communicable diseases, including proper nutrition and sufficient physical activity, is undoubtedly of great importance, especially for the able-bodied population. The results of the research presented in the manuscript are of particular interest to health care organizers.

However, I have a number of questions and comments:

1. The title of the article requires correction, as it does not correspond to the purpose or objectives of the study.

2. As the authors point out, the purpose of the study was to study the factors influencing the possibility of obtaining educational information through messages to the respondents' personal e-mail. Neither the initial level of awareness of the presence of NCDs risk factors, proper nutrition, proper level of physical activity, nor the dynamics of these parameters after exposure were analyzed. It is not clear on what basis the authors draw conclusions about the positive effect of e-mail messages on the change in eating behavior and physical activity of respondents.

3. The conclusion about the positive effect of mail messages on the reduction of sick leave is also doubtful.

4. The conclusion about the greater involvement and use of the Internet, e-mail by workers with a sedentary type of work is logical, compared with workers with mostly heavy physical labor and standing work, since it is unlikely that these workers have access to the Internet and mail during the performance of such work.

5. The conclusion about the higher frequency of involvement in the educational process is also questionable only on the basis of information about the letters read. It is necessary to obtain information about the dynamics of the level of awareness and commitment of respondents to preventive measures.

6. Figures 7 and 8 are not presented in the manuscript.

I think the authors need to finalize the manuscript and supplement the missing research results.

Comments on the Quality of English Language

Moderate editing of English language required

Author Response

Our responses to each point raised by the Reviewer 2 are marked with arrows below the comments (à). Changes in the manuscript are marked in red.

We trust that we have adequately answered all the comments and made appropriate changes. We are grateful for the constructive recommendations that resulted in an improved text and a better understanding of the revised manuscript.

In conclusion, we hope that the revised manuscript is now acceptable for publication in your respected journal. Thank you for your attention and interest.

Reviewers' comments 2:

  1. The title of the article requires correction, as it does not correspond to the purpose or objectives of the study.

à As requested by the Reviewer 2 (and commented by the Reviewer 1), the present manuscript was reviewed and the title was correspondingly corrected. The new title is: Engagement frequency in web-based nutrition and physical activity educational campaign.

  1. As the authors point out, the purpose of the study was to study the factors influencing the possibility of obtaining educational information through messages to the respondents' personal e-mail. Neither the initial level of awareness of the presence of NCDs risk factors, proper nutrition, proper level of physical activity, nor the dynamics of these parameters after exposure were analyzed. It is not clear on what basis the authors draw conclusions about the positive effect of e-mail messages on the change in eating behavior and physical activity of respondents.

à In connection with the comment, we added a limitation with explanation: The study acknowledges several limitations that warrant consideration. The utilization of a descriptive, cross-sectional, correlational design constrains our ability to make definitive assertions about cause–effect relationships. Our research design may not have accounted for all relevant factors, potentially introducing uncontrolled variables. Consequently, the correlations observed should not be interpreted as implying causation, as confounding factors (such as education) may influence the observed relationships. Furthermore, despite the study spanning a 2-year period, there is a possibility that dynamic processes were overlooked.

  1. The conclusion about the positive effect of mail messages on the reduction of sick leave is also doubtful.

à We addressed this issue in study limitations and have carefully checked the text in order to omit any parts of the text that might imply observation or detection of causal relationships.

  1. The conclusion about the greater involvement and use of the Internet, e-mail by workers with a sedentary type of work is logical, compared with workers with mostly heavy physical labor and standing work, since it is unlikely that these workers have access to the Internet and mail during the performance of such work.

à Thank you for raising a valid and relevant point. To enhance the clarity of our reasoning and the foundation of our conclusion, we have explicitly stated in the manuscript that, throughout the project/research, we asked respondents to provide email addresses that they access daily, preferably through their mobile phones. Consequently, the majority of participants provided us with their personal email accounts. This ensures that the content was available to the vast majority of participants both during and outside of work hours, independent of Internet access availability during work hours.

  1. The conclusion about the higher frequency of involvement in the educational process is also questionable only on the basis of information about the letters read. It is necessary to obtain information about the dynamics of the level of awareness and commitment of respondents to preventive measures.

à This is a valid observation for which we added an additional explanation to support our inference. We rewrote the methods with detailed information on the operationalization of 'Email opened = Email read.' Each email sent during the study contained a link to content on a landing page, which, in turn, included further links to content or video. At the conclusion of each landing page, participants were prompted to provide a content rating. We established that an email was 'read' only for those individuals who actively gave a content rating. Importantly, it is acknowledged that this operationalization does not confirm whether the email was read and considered in full. We delve into this aspect further in the limitations section of the study.

  1. Figures 7 and 8 are not presented in the manuscript.

à We sincerely apologize for the oversight regarding the absence of two figures in the uploaded manuscript on the journal's website. Unfortunately, we currently do not have a clear explanation for this discrepancy, as the version on our local PCs includes the mentioned figures. We understand the importance of a comprehensive submission and deeply regret any inconvenience this may have caused.

  1. I think the authors need to finalize the manuscript and supplement the missing research results.

à Please see the previous point. In addition, the entire text in the article has been supplemented with suggestions from Reviewer 1 and 2. Additions were made in the abstract, methods, results, discussion, and conclusion sections.

Round 2

Reviewer 1 Report

Comments and Suggestions for Authors

Reviewers' comments 1:

Thank you very much for your detailed responses and the adjustments you have made, as well as the additional information provided. However, I still have the following concerns:

  1. It is impressive that the study analyzes data over a period of 5 years. However, measuring the impact on NCDs within this timeframe may be challenging. NCDs arise from long-term unfavorable life and lifestyle conditions. Five years are relatively short in this context, raising the question of whether meaningful associations can be measured?

à As advised by Reviewer 1, we changed the text and removed the sentence that suggests a connection. We also add the sentence: The entire project started on 1 September 2017 and was completed on 30 September 2022, and the campaign in the field of nutrition and physical activity lasted from 2020 to 2022.

I appreciate your response. However, my concern persists, and it becomes even more pronounced if the study duration is limited to two years instead of five. Measuring the impact on NCDs within this timeframe might pose challenges, raising questions about the feasibility of detecting meaningful associations. At the end of the introduction, you mentioned that one of the objectives of the study was to examine differences in absenteeism due to NCDs between the SMEs included and those not included in the web-based educational campaigns. I would like to suggest to rather focus on evaluating the impact on behaviors associated with NCDs.

  1. In the methods and research questions, it appears that the correlation with sick leave and absence days is an important central question of the study. I indeed find this to be a crucial aspect, as it represents an objective indicator. However, in the results, it is only presented very briefly and lacks differentiation.

à The utilization of a descriptive, cross-sectional, correlational design constrains our ability to make definitive assertions about cause–effect relationships. Our research design may not have accounted for all relevant factors, potentially introducing uncontrolled variables. Consequently, the correlations observed should not be interpreted as implying causation, as confounding factors (such as education) may influence the observed relationships. Furthermore, despite the study spanning a 2-year period, there is a possibility that dynamic processes were overlooked.

Thank you for your response. However, my intention was not to inquire about the capacity to draw cause-and-effect conclusions. Given that one of your stated objectives is to explore differences in absenteeism due to NCDs, I was anticipating a more detailed analysis on this aspect. The current presentation suggests that this is a primary objective of the paper.

  1. Importantly, were baseline differences between participating and non-participating companies controlled for? Otherwise, drawing conclusions might be difficult.

àTo address the reviewer's observation regarding the need to control for differences between participating and non-participating companies, it's important to note that our research design did not include a control group. We have taken steps to clarify this aspect by extensively revising the Research Limitations section, explicitly stating that the design did not envisage the inclusion of a control group. This information is now articulated in a more detailed manner to provide a clear understanding of the study's limitations. We appreciate the reviewer's attention to this matter and believe that the revised section adequately addresses the raised concern.

Thank you for your response. You mentioned that your design did not incorporate a control group. However, one of the study objectives is to "examine the differences in absenteeism due to NCDs between the SMEs included and those not included in the web-based educational campaigns." The results section includes an analysis of the duration of sick leave over time and between participating and non-participating companies. Consequently, I still have concerns about potential baseline differences between these groups.

Thank you for including Figure 7 and Figure 8, as they help to understand the reported results. Nevertheless, I am curious to know if you conducted statistical analyses to assess changes over time and differences between companies, or if these are purely observational findings. You mentioned that sick leave did not decrease in non-participating companies but increased on average. Could you please provide the results of statistical tests to support these observations?

Discussion:

  1. In the discussion, it is stated, "We developed web-based content on health eating and PA." ? Reading the methods, I understood that only an existing platform was used. Please specify.

àAs requested by the Reviewer 1, the Discussion section of the revised manuscript now contains additional specifications, where we rewrote the text, more specifically conveying that both platform and the content were newly developed.

Unfortunately, I was not able to find the respective paragraph in the discussion section. If you developed and designed the platform and content specifically for this study, it would be important to highlight this aspect in the methods section. Please provide specific details in the methods section to facilitate a comprehensive understanding of your study and its content.

  1. I miss in the discussion the question of whether the results might also be explained by the fact that individuals engaged in sedentary work can more easily read emails during their workday.

à Thank you for raising a valid and relevant point. To enhance the clarity of our reasoning and the foundation of our conclusion, we have explicitly stated in the manuscript that, throughout the project/research, we asked respondents to provide email addresses that they access daily, preferably through their mobile phones. Consequently, the majority of participants provided us with their personal email accounts. This ensures that the content was available to the vast majority of participants both during and outside of work hours, independent of Internet access availability during work hours. Of course, there is still the possibility of the mentioned influence, which we wrote in the limitations.

Thank you for providing clarification that the emails were sent to personal email accounts. However, I would like to suggest considering the possibility that individuals in sedentary roles, such as those working at a computer, may have greater accessibility to read emails during their workday compared to those involved in more physically demanding tasks. It would be valuable to know when individuals accessed the emails and the associated information—whether it occurred within or outside their working hours. And did this differ between groups?

  1. Further, to what extent are those differences possibly explained by educational level? Are individuals with sedentary jobs more educated than those with physical work? Was this controlled for? Please specify.

à It is possible, because it has been proven that education affects the eating habits. But we must take into account that in companies in question, mainly due to their small size, the structure of employees is very heterogeneous in terms of the tasks they perform, but not so much in terms of education. The vast majority of employees in the sample have secondary and higher education, which reduces the possibility of these differences. We included this reasoning in the discussion.

Thank you for including those considerations in your discussion section. However, please provide sources for statements such as “it has been proven that education and personal income affects the eating habits” how and by whom? It is well-known that … Please incorporate your rationale within the existing literature providing links to relevant studies and research.

Author Response

Our responses to each point raised by the Reviewer 1 are written in green font, added/changed text in the article is made by tracking the changes in the text.

Thank you very much for your detailed responses and the adjustments you have made, as well as the additional information provided. However, I still have the following concerns:

  1. It is impressive that the study analyzes data over a period of 5 years. However, measuring the impact on NCDs within this timeframe may be challenging. NCDs arise from long-term unfavorable life and lifestyle conditions. Five years are relatively short in this context, raising the question of whether meaningful associations can be measured?

As advised by Reviewer 1, we changed the text and removed the sentence that suggests a connection. We also add the sentence: The entire project started on 1 September 2017 and was completed on 30 September 2022, and the campaign in the field of nutrition and physical activity lasted from 2020 to 2022.

I appreciate your response. However, my concern persists, and it becomes even more pronounced if the study duration is limited to two years instead of five. Measuring the impact on NCDs within this timeframe might pose challenges, raising questions about the feasibility of detecting meaningful associations. At the end of the introduction, you mentioned that one of the objectives of the study was to examine differences in absenteeism due to NCDs between the SMEs included and those not included in the web-based educational campaigns. I would like to suggest to rather focus on evaluating the impact on behaviors associated with NCDs.

Thank you for your thoughtful and considered feedback. We take due note of your concerns regarding the study duration, particularly the potential challenges in measuring the impact on NCDs within a two-year timeframe. We recognise the importance of collecting data and we have carefully considered the timeframe of the study. We have re-examined the data and found it to be of insufficient quality and usefulness. For this reason, we have excluded all text relating to absenteeism due to NCDs from the article. We are grateful for your suggestion as it adds to the quality and relevance of our research.

  1. In the methods and research questions, it appears that the correlation with sick leave and absence days is an important central question of the study. I indeed find this to be a crucial aspect, as it represents an objective indicator. However, in the results, it is only presented very briefly and lacks differentiation.

The utilization of a descriptive, cross-sectional, correlational design constrains our ability to make definitive assertions about cause–effect relationships. Our research design may not have accounted for all relevant factors, potentially introducing uncontrolled variables. Consequently, the correlations observed should not be interpreted as implying causation, as confounding factors (such as education) may influence the observed relationships. Furthermore, despite the study spanning a 2-year period, there is a possibility that dynamic processes were overlooked.

Thank you for your response. However, my intention was not to inquire about the capacity to draw cause-and-effect conclusions. Given that one of your stated objectives is to explore differences in absenteeism due to NCDs, I was anticipating a more detailed analysis on this aspect. The current presentation suggests that this is a primary objective of the paper.

We sincerely apologize for any confusion caused. In our effort to refine the article, we have taken the initiative to remove sections that may have unintentionally conveyed a misleading emphasis on the difference in absenteeism due to NCDs as the primary objective of the paper.

  1. Importantly, were baseline differences between participating and non-participating companies controlled for? Otherwise, drawing conclusions might be difficult.

To address the reviewer's observation regarding the need to control for differences between participating and non-participating companies, it's important to note that our research design did not include a control group. We have taken steps to clarify this aspect by extensively revising the Research Limitations section, explicitly stating that the design did not envisage the inclusion of a control group. This information is now articulated in a more detailed manner to provide a clear understanding of the study's limitations. We appreciate the reviewer's attention to this matter and believe that the revised section adequately addresses the raised concern.

Thank you for your response. You mentioned that your design did not incorporate a control group. However, one of the study objectives is to "examine the differences in absenteeism due to NCDs between the SMEs included and those not included in the web-based educational campaigns." The results section includes an analysis of the duration of sick leave over time and between participating and non-participating companies. Consequently, I still have concerns about potential baseline differences between these groups.

Thank you for including Figure 7 and Figure 8, as they help to understand the reported results. Nevertheless, I am curious to know if you conducted statistical analyses to assess changes over time and differences between companies, or if these are purely observational findings. You mentioned that sick leave did not decrease in non-participating companies but increased on average. Could you please provide the results of statistical tests to support these observations?

Thank you for thoughtful persistence in pointing out the problems related to NCDs. As already mentioned, upon careful consideration, we made the decision to remove the tables from the article due to the data's insufficient quality for conducting rigorous statistical tests and discerning differences between participating and non-participating companies. The updated version of the article no longer includes the text and comparisons related to absenteeism due to NCDs.

Discussion:

  1. In the discussion, it is stated, "We developed web-based content on health eating and PA." ? Reading the methods, I understood that only an existing platform was used. Please specify.

As requested by the Reviewer 1, the Discussion section of the revised manuscript now contains additional specifications, where we rewrote the text, more specifically conveying that both platform and the content were newly developed.

Unfortunately, I was not able to find the respective paragraph in the discussion section. If you developed and designed the platform and content specifically for this study, it would be important to highlight this aspect in the methods section. Please provide specific details in the methods section to facilitate a comprehensive understanding of your study and its content.

As requested by Reviewer 1, the Methods in 2.4. section of the revised manuscript now contains additional information on web-based content: The e-content in forms short text supplemented by links to articles of was collaboratively created by a dietitian, a nutritionist, and a physician, providing a comprehensive understanding of the benefits of a healthy diet. Information on topics such as obesity, sugar consumption, benefits of breakfast, vitamins (B and D), healthy foods, nutrition for a healthy heart, colon cancer and nutrition, as well as safe foods on vacation, was provided by the physician. A poster highlighting the importance of healthy foods was designed by the dietitian and nutritionist. E-materials on food rhythms, including text and quizzes for awareness, and energy requirement determination [25] through personalized calculations were also facilitated. For practical application, guidelines for planning balanced meals in various situations, offering twenty recipes and video recordings, were provided by the nutritionist. Effective portion size measurement was succinctly described in a short article and on a poster as well as the emphasis on fruit and vegetable consumption accompanied with quick tips on assembling a balanced meal. The importance of adequate fluid intake was also stressed, providing recipes, video recordings for drink preparation, and posters. Addressing salt and spice consumption, information on spice benefits, presentations of various spices, and ways to reduce salt intake was offered. Content on specific diets, such as the Mediterranean diet, including texts and quizzes, was also prepared by dietitians and nutritionists, as well as the concept of healthy eating for the planet. All physical activity materials, overseen by an experienced kinesiologist and physiotherapist, featured a text followed by ten videos demonstrating correct exercise execution for various activities in different occupations.

The information pertaining to the design of the e-platform is provided in section 2.2.

  1. I miss in the discussion the question of whether the results might also be explained by the fact that individuals engaged in sedentary work can more easily read emails during their workday.

Thank you for raising a valid and relevant point. To enhance the clarity of our reasoning and the foundation of our conclusion, we have explicitly stated in the manuscript that, throughout the project/research, we asked respondents to provide email addresses that they access daily, preferably through their mobile phones. Consequently, the majority of participants provided us with their personal email accounts. This ensures that the content was available to the vast majority of participants both during and outside of work hours, independent of Internet access availability during work hours. Of course, there is still the possibility of the mentioned influence, which we wrote in the limitations.

Thank you for providing clarification that the emails were sent to personal email accounts. However, I would like to suggest considering the possibility that individuals in sedentary roles, such as those working at a computer, may have greater accessibility to read emails during their workday compared to those involved in more physically demanding tasks.

We would like to thank you for your valuable suggestion. Following the recommendation, we have included a sentence in the discussion section to provide a clearer perspective on the potential accessibility of sedentary workers to read emails during the working day, compared to workers with more physically demanding tasks: It's worth contemplating the notion that individuals in sedentary roles, particularly those working at a computer, might have enhanced accessibility to read emails throughout their workday, distinguishing them from individuals engaged in more physically demanding tasks. This consideration could shed light on potential variations in email engagement based on the nature of job responsibilities.

It would be valuable to know when individuals accessed the emails and the associated information—whether it occurred within or outside their working hours. And did this differ between groups?

Yes, that would certainly be interesting. Unfortunately, it is a challenge to determine whether employees have accessed the content during their working hours or in their free time. While our CRM system can record the IP address, identify the device (phone, tablet, or computer) used to access the campaign, and note the time of access, it does not provide data that would conclusively distinguish between work and leisure time.

  1. Further, to what extent are those differences possibly explained by educational level? Are individuals with sedentary jobs more educated than those with physical work? Was this controlled for? Please specify.

It is possible, because it has been proven that education affects the eating habits. But we must take into account that in companies in question, mainly due to their small size, the structure of employees is very heterogeneous in terms of the tasks they perform, but not so much in terms of education. The vast majority of employees in the sample have secondary and higher education, which reduces the possibility of these differences. We included this reasoning in the discussion.

Thank you for including those considerations in your discussion section. However, please provide sources for statements such as “it has been proven that education and personal income affects the eating habits” how and by whom? It is well-known that … Please incorporate your rationale within the existing literature providing links to relevant studies and research.

As suggested by the reviewer, we have added relevant literature:

It has been proven that education and personal income affects the eating habits. According to a Slovenian study, men with a low level of education and low income have the worst diet. They skip breakfast, eat too much red meat and consume too much saturated fat [32].

  1. National Institute of Public Health, Slovenia. Different aspects of eating habits of inhabitants of Slovenia. Available online in Slovenian language: https://nijz.si/wp-content/uploads/2022/07/razlicni_vidiki_prehranjevanja_prebivalcev_slovenije.pdf (accessed on 5 Dec. 2023)

It is well known that workers in more physically active occupations and lower education and income (truck drivers, construction workers, etc.) is “hard-to-reach” with digital health interventions, especially male, which is also apply to this research [33, 34].

  1. Rasmussen, C. L.; Dumuid, D.; Hron, K. Day-to-day pattern of work and leisure time physical behaviours: are low socioeconomic status adults couch potatoes or work warriors?. BMC Public Health. 2021, 21, 1342. doi:10.1186/s12889-021-11409-0
  2. Sendall, M. C.; McCosker, L. K.; Crane, P.; Rowland, B.; Fleming, M.; Biggs, H. C. Using Facebook for Health Promotion in "Hard-to-Reach" Truck Drivers: Qualitative Analysis. J Med Internet Res. 2018, 20, e286. doi:10.2196/jmir.9689

    Our responses to each point raised by the Reviewer 1 are written in green font, added/changed text in the article is made by tracking the changes in the text.

    Thank you very much for your detailed responses and the adjustments you have made, as well as the additional information provided. However, I still have the following concerns:

    1. It is impressive that the study analyzes data over a period of 5 years. However, measuring the impact on NCDs within this timeframe may be challenging. NCDs arise from long-term unfavorable life and lifestyle conditions. Five years are relatively short in this context, raising the question of whether meaningful associations can be measured?

    As advised by Reviewer 1, we changed the text and removed the sentence that suggests a connection. We also add the sentence: The entire project started on 1 September 2017 and was completed on 30 September 2022, and the campaign in the field of nutrition and physical activity lasted from 2020 to 2022.

    I appreciate your response. However, my concern persists, and it becomes even more pronounced if the study duration is limited to two years instead of five. Measuring the impact on NCDs within this timeframe might pose challenges, raising questions about the feasibility of detecting meaningful associations. At the end of the introduction, you mentioned that one of the objectives of the study was to examine differences in absenteeism due to NCDs between the SMEs included and those not included in the web-based educational campaigns. I would like to suggest to rather focus on evaluating the impact on behaviors associated with NCDs.

    Thank you for your thoughtful and considered feedback. We take due note of your concerns regarding the study duration, particularly the potential challenges in measuring the impact on NCDs within a two-year timeframe. We recognise the importance of collecting data and we have carefully considered the timeframe of the study. We have re-examined the data and found it to be of insufficient quality and usefulness. For this reason, we have excluded all text relating to absenteeism due to NCDs from the article. We are grateful for your suggestion as it adds to the quality and relevance of our research.

    1. In the methods and research questions, it appears that the correlation with sick leave and absence days is an important central question of the study. I indeed find this to be a crucial aspect, as it represents an objective indicator. However, in the results, it is only presented very briefly and lacks differentiation.

    The utilization of a descriptive, cross-sectional, correlational design constrains our ability to make definitive assertions about cause–effect relationships. Our research design may not have accounted for all relevant factors, potentially introducing uncontrolled variables. Consequently, the correlations observed should not be interpreted as implying causation, as confounding factors (such as education) may influence the observed relationships. Furthermore, despite the study spanning a 2-year period, there is a possibility that dynamic processes were overlooked.

    Thank you for your response. However, my intention was not to inquire about the capacity to draw cause-and-effect conclusions. Given that one of your stated objectives is to explore differences in absenteeism due to NCDs, I was anticipating a more detailed analysis on this aspect. The current presentation suggests that this is a primary objective of the paper.

    We sincerely apologize for any confusion caused. In our effort to refine the article, we have taken the initiative to remove sections that may have unintentionally conveyed a misleading emphasis on the difference in absenteeism due to NCDs as the primary objective of the paper.

    1. Importantly, were baseline differences between participating and non-participating companies controlled for? Otherwise, drawing conclusions might be difficult.

    To address the reviewer's observation regarding the need to control for differences between participating and non-participating companies, it's important to note that our research design did not include a control group. We have taken steps to clarify this aspect by extensively revising the Research Limitations section, explicitly stating that the design did not envisage the inclusion of a control group. This information is now articulated in a more detailed manner to provide a clear understanding of the study's limitations. We appreciate the reviewer's attention to this matter and believe that the revised section adequately addresses the raised concern.

    Thank you for your response. You mentioned that your design did not incorporate a control group. However, one of the study objectives is to "examine the differences in absenteeism due to NCDs between the SMEs included and those not included in the web-based educational campaigns." The results section includes an analysis of the duration of sick leave over time and between participating and non-participating companies. Consequently, I still have concerns about potential baseline differences between these groups.

    Thank you for including Figure 7 and Figure 8, as they help to understand the reported results. Nevertheless, I am curious to know if you conducted statistical analyses to assess changes over time and differences between companies, or if these are purely observational findings. You mentioned that sick leave did not decrease in non-participating companies but increased on average. Could you please provide the results of statistical tests to support these observations?

    Thank you for thoughtful persistence in pointing out the problems related to NCDs. As already mentioned, upon careful consideration, we made the decision to remove the tables from the article due to the data's insufficient quality for conducting rigorous statistical tests and discerning differences between participating and non-participating companies. The updated version of the article no longer includes the text and comparisons related to absenteeism due to NCDs.

    Discussion:

    1. In the discussion, it is stated, "We developed web-based content on health eating and PA." ? Reading the methods, I understood that only an existing platform was used. Please specify.

    As requested by the Reviewer 1, the Discussion section of the revised manuscript now contains additional specifications, where we rewrote the text, more specifically conveying that both platform and the content were newly developed.

    Unfortunately, I was not able to find the respective paragraph in the discussion section. If you developed and designed the platform and content specifically for this study, it would be important to highlight this aspect in the methods section. Please provide specific details in the methods section to facilitate a comprehensive understanding of your study and its content.

    As requested by Reviewer 1, the Methods in 2.4. section of the revised manuscript now contains additional information on web-based content: The e-content in forms short text supplemented by links to articles of was collaboratively created by a dietitian, a nutritionist, and a physician, providing a comprehensive understanding of the benefits of a healthy diet. Information on topics such as obesity, sugar consumption, benefits of breakfast, vitamins (B and D), healthy foods, nutrition for a healthy heart, colon cancer and nutrition, as well as safe foods on vacation, was provided by the physician. A poster highlighting the importance of healthy foods was designed by the dietitian and nutritionist. E-materials on food rhythms, including text and quizzes for awareness, and energy requirement determination [25] through personalized calculations were also facilitated. For practical application, guidelines for planning balanced meals in various situations, offering twenty recipes and video recordings, were provided by the nutritionist. Effective portion size measurement was succinctly described in a short article and on a poster as well as the emphasis on fruit and vegetable consumption accompanied with quick tips on assembling a balanced meal. The importance of adequate fluid intake was also stressed, providing recipes, video recordings for drink preparation, and posters. Addressing salt and spice consumption, information on spice benefits, presentations of various spices, and ways to reduce salt intake was offered. Content on specific diets, such as the Mediterranean diet, including texts and quizzes, was also prepared by dietitians and nutritionists, as well as the concept of healthy eating for the planet. All physical activity materials, overseen by an experienced kinesiologist and physiotherapist, featured a text followed by ten videos demonstrating correct exercise execution for various activities in different occupations.

    The information pertaining to the design of the e-platform is provided in section 2.2.

    1. I miss in the discussion the question of whether the results might also be explained by the fact that individuals engaged in sedentary work can more easily read emails during their workday.

    Thank you for raising a valid and relevant point. To enhance the clarity of our reasoning and the foundation of our conclusion, we have explicitly stated in the manuscript that, throughout the project/research, we asked respondents to provide email addresses that they access daily, preferably through their mobile phones. Consequently, the majority of participants provided us with their personal email accounts. This ensures that the content was available to the vast majority of participants both during and outside of work hours, independent of Internet access availability during work hours. Of course, there is still the possibility of the mentioned influence, which we wrote in the limitations.

    Thank you for providing clarification that the emails were sent to personal email accounts. However, I would like to suggest considering the possibility that individuals in sedentary roles, such as those working at a computer, may have greater accessibility to read emails during their workday compared to those involved in more physically demanding tasks.

    We would like to thank you for your valuable suggestion. Following the recommendation, we have included a sentence in the discussion section to provide a clearer perspective on the potential accessibility of sedentary workers to read emails during the working day, compared to workers with more physically demanding tasks: It's worth contemplating the notion that individuals in sedentary roles, particularly those working at a computer, might have enhanced accessibility to read emails throughout their workday, distinguishing them from individuals engaged in more physically demanding tasks. This consideration could shed light on potential variations in email engagement based on the nature of job responsibilities.

    It would be valuable to know when individuals accessed the emails and the associated information—whether it occurred within or outside their working hours. And did this differ between groups?

    Yes, that would certainly be interesting. Unfortunately, it is a challenge to determine whether employees have accessed the content during their working hours or in their free time. While our CRM system can record the IP address, identify the device (phone, tablet, or computer) used to access the campaign, and note the time of access, it does not provide data that would conclusively distinguish between work and leisure time.

    1. Further, to what extent are those differences possibly explained by educational level? Are individuals with sedentary jobs more educated than those with physical work? Was this controlled for? Please specify.

    It is possible, because it has been proven that education affects the eating habits. But we must take into account that in companies in question, mainly due to their small size, the structure of employees is very heterogeneous in terms of the tasks they perform, but not so much in terms of education. The vast majority of employees in the sample have secondary and higher education, which reduces the possibility of these differences. We included this reasoning in the discussion.

    Thank you for including those considerations in your discussion section. However, please provide sources for statements such as “it has been proven that education and personal income affects the eating habits” how and by whom? It is well-known that … Please incorporate your rationale within the existing literature providing links to relevant studies and research.

    As suggested by the reviewer, we have added relevant literature:

    It has been proven that education and personal income affects the eating habits. According to a Slovenian study, men with a low level of education and low income have the worst diet. They skip breakfast, eat too much red meat and consume too much saturated fat [32].

    1. National Institute of Public Health, Slovenia. Different aspects of eating habits of inhabitants of Slovenia. Available online in Slovenian language: https://nijz.si/wp-content/uploads/2022/07/razlicni_vidiki_prehranjevanja_prebivalcev_slovenije.pdf (accessed on 5 Dec. 2023)

    It is well known that workers in more physically active occupations and lower education and income (truck drivers, construction workers, etc.) is “hard-to-reach” with digital health interventions, especially male, which is also apply to this research [33, 34].

    1. Rasmussen, C. L.; Dumuid, D.; Hron, K. Day-to-day pattern of work and leisure time physical behaviours: are low socioeconomic status adults couch potatoes or work warriors?. BMC Public Health. 2021, 21, 1342. doi:10.1186/s12889-021-11409-0
    2. Sendall, M. C.; McCosker, L. K.; Crane, P.; Rowland, B.; Fleming, M.; Biggs, H. C. Using Facebook for Health Promotion in "Hard-to-Reach" Truck Drivers: Qualitative Analysis. J Med Internet Res. 2018, 20, e286. doi:10.2196/jmir.9689

      Our responses to each point raised by the Reviewer 1 are written in green font, added/changed text in the article is made by tracking the changes in the text.

      Thank you very much for your detailed responses and the adjustments you have made, as well as the additional information provided. However, I still have the following concerns:

      1. It is impressive that the study analyzes data over a period of 5 years. However, measuring the impact on NCDs within this timeframe may be challenging. NCDs arise from long-term unfavorable life and lifestyle conditions. Five years are relatively short in this context, raising the question of whether meaningful associations can be measured?

      As advised by Reviewer 1, we changed the text and removed the sentence that suggests a connection. We also add the sentence: The entire project started on 1 September 2017 and was completed on 30 September 2022, and the campaign in the field of nutrition and physical activity lasted from 2020 to 2022.

      I appreciate your response. However, my concern persists, and it becomes even more pronounced if the study duration is limited to two years instead of five. Measuring the impact on NCDs within this timeframe might pose challenges, raising questions about the feasibility of detecting meaningful associations. At the end of the introduction, you mentioned that one of the objectives of the study was to examine differences in absenteeism due to NCDs between the SMEs included and those not included in the web-based educational campaigns. I would like to suggest to rather focus on evaluating the impact on behaviors associated with NCDs.

      Thank you for your thoughtful and considered feedback. We take due note of your concerns regarding the study duration, particularly the potential challenges in measuring the impact on NCDs within a two-year timeframe. We recognise the importance of collecting data and we have carefully considered the timeframe of the study. We have re-examined the data and found it to be of insufficient quality and usefulness. For this reason, we have excluded all text relating to absenteeism due to NCDs from the article. We are grateful for your suggestion as it adds to the quality and relevance of our research.

      1. In the methods and research questions, it appears that the correlation with sick leave and absence days is an important central question of the study. I indeed find this to be a crucial aspect, as it represents an objective indicator. However, in the results, it is only presented very briefly and lacks differentiation.

      The utilization of a descriptive, cross-sectional, correlational design constrains our ability to make definitive assertions about cause–effect relationships. Our research design may not have accounted for all relevant factors, potentially introducing uncontrolled variables. Consequently, the correlations observed should not be interpreted as implying causation, as confounding factors (such as education) may influence the observed relationships. Furthermore, despite the study spanning a 2-year period, there is a possibility that dynamic processes were overlooked.

      Thank you for your response. However, my intention was not to inquire about the capacity to draw cause-and-effect conclusions. Given that one of your stated objectives is to explore differences in absenteeism due to NCDs, I was anticipating a more detailed analysis on this aspect. The current presentation suggests that this is a primary objective of the paper.

      We sincerely apologize for any confusion caused. In our effort to refine the article, we have taken the initiative to remove sections that may have unintentionally conveyed a misleading emphasis on the difference in absenteeism due to NCDs as the primary objective of the paper.

      1. Importantly, were baseline differences between participating and non-participating companies controlled for? Otherwise, drawing conclusions might be difficult.

      To address the reviewer's observation regarding the need to control for differences between participating and non-participating companies, it's important to note that our research design did not include a control group. We have taken steps to clarify this aspect by extensively revising the Research Limitations section, explicitly stating that the design did not envisage the inclusion of a control group. This information is now articulated in a more detailed manner to provide a clear understanding of the study's limitations. We appreciate the reviewer's attention to this matter and believe that the revised section adequately addresses the raised concern.

      Thank you for your response. You mentioned that your design did not incorporate a control group. However, one of the study objectives is to "examine the differences in absenteeism due to NCDs between the SMEs included and those not included in the web-based educational campaigns." The results section includes an analysis of the duration of sick leave over time and between participating and non-participating companies. Consequently, I still have concerns about potential baseline differences between these groups.

      Thank you for including Figure 7 and Figure 8, as they help to understand the reported results. Nevertheless, I am curious to know if you conducted statistical analyses to assess changes over time and differences between companies, or if these are purely observational findings. You mentioned that sick leave did not decrease in non-participating companies but increased on average. Could you please provide the results of statistical tests to support these observations?

      Thank you for thoughtful persistence in pointing out the problems related to NCDs. As already mentioned, upon careful consideration, we made the decision to remove the tables from the article due to the data's insufficient quality for conducting rigorous statistical tests and discerning differences between participating and non-participating companies. The updated version of the article no longer includes the text and comparisons related to absenteeism due to NCDs.

      Discussion:

      1. In the discussion, it is stated, "We developed web-based content on health eating and PA." ? Reading the methods, I understood that only an existing platform was used. Please specify.

      As requested by the Reviewer 1, the Discussion section of the revised manuscript now contains additional specifications, where we rewrote the text, more specifically conveying that both platform and the content were newly developed.

      Unfortunately, I was not able to find the respective paragraph in the discussion section. If you developed and designed the platform and content specifically for this study, it would be important to highlight this aspect in the methods section. Please provide specific details in the methods section to facilitate a comprehensive understanding of your study and its content.

      As requested by Reviewer 1, the Methods in 2.4. section of the revised manuscript now contains additional information on web-based content: The e-content in forms short text supplemented by links to articles of was collaboratively created by a dietitian, a nutritionist, and a physician, providing a comprehensive understanding of the benefits of a healthy diet. Information on topics such as obesity, sugar consumption, benefits of breakfast, vitamins (B and D), healthy foods, nutrition for a healthy heart, colon cancer and nutrition, as well as safe foods on vacation, was provided by the physician. A poster highlighting the importance of healthy foods was designed by the dietitian and nutritionist. E-materials on food rhythms, including text and quizzes for awareness, and energy requirement determination [25] through personalized calculations were also facilitated. For practical application, guidelines for planning balanced meals in various situations, offering twenty recipes and video recordings, were provided by the nutritionist. Effective portion size measurement was succinctly described in a short article and on a poster as well as the emphasis on fruit and vegetable consumption accompanied with quick tips on assembling a balanced meal. The importance of adequate fluid intake was also stressed, providing recipes, video recordings for drink preparation, and posters. Addressing salt and spice consumption, information on spice benefits, presentations of various spices, and ways to reduce salt intake was offered. Content on specific diets, such as the Mediterranean diet, including texts and quizzes, was also prepared by dietitians and nutritionists, as well as the concept of healthy eating for the planet. All physical activity materials, overseen by an experienced kinesiologist and physiotherapist, featured a text followed by ten videos demonstrating correct exercise execution for various activities in different occupations.

      The information pertaining to the design of the e-platform is provided in section 2.2.

      1. I miss in the discussion the question of whether the results might also be explained by the fact that individuals engaged in sedentary work can more easily read emails during their workday.

      Thank you for raising a valid and relevant point. To enhance the clarity of our reasoning and the foundation of our conclusion, we have explicitly stated in the manuscript that, throughout the project/research, we asked respondents to provide email addresses that they access daily, preferably through their mobile phones. Consequently, the majority of participants provided us with their personal email accounts. This ensures that the content was available to the vast majority of participants both during and outside of work hours, independent of Internet access availability during work hours. Of course, there is still the possibility of the mentioned influence, which we wrote in the limitations.

      Thank you for providing clarification that the emails were sent to personal email accounts. However, I would like to suggest considering the possibility that individuals in sedentary roles, such as those working at a computer, may have greater accessibility to read emails during their workday compared to those involved in more physically demanding tasks.

      We would like to thank you for your valuable suggestion. Following the recommendation, we have included a sentence in the discussion section to provide a clearer perspective on the potential accessibility of sedentary workers to read emails during the working day, compared to workers with more physically demanding tasks: It's worth contemplating the notion that individuals in sedentary roles, particularly those working at a computer, might have enhanced accessibility to read emails throughout their workday, distinguishing them from individuals engaged in more physically demanding tasks. This consideration could shed light on potential variations in email engagement based on the nature of job responsibilities.

      It would be valuable to know when individuals accessed the emails and the associated information—whether it occurred within or outside their working hours. And did this differ between groups?

      Yes, that would certainly be interesting. Unfortunately, it is a challenge to determine whether employees have accessed the content during their working hours or in their free time. While our CRM system can record the IP address, identify the device (phone, tablet, or computer) used to access the campaign, and note the time of access, it does not provide data that would conclusively distinguish between work and leisure time.

      1. Further, to what extent are those differences possibly explained by educational level? Are individuals with sedentary jobs more educated than those with physical work? Was this controlled for? Please specify.

      It is possible, because it has been proven that education affects the eating habits. But we must take into account that in companies in question, mainly due to their small size, the structure of employees is very heterogeneous in terms of the tasks they perform, but not so much in terms of education. The vast majority of employees in the sample have secondary and higher education, which reduces the possibility of these differences. We included this reasoning in the discussion.

      Thank you for including those considerations in your discussion section. However, please provide sources for statements such as “it has been proven that education and personal income affects the eating habits” how and by whom? It is well-known that … Please incorporate your rationale within the existing literature providing links to relevant studies and research.

      As suggested by the reviewer, we have added relevant literature:

      It has been proven that education and personal income affects the eating habits. According to a Slovenian study, men with a low level of education and low income have the worst diet. They skip breakfast, eat too much red meat and consume too much saturated fat [32].

      1. National Institute of Public Health, Slovenia. Different aspects of eating habits of inhabitants of Slovenia. Available online in Slovenian language: https://nijz.si/wp-content/uploads/2022/07/razlicni_vidiki_prehranjevanja_prebivalcev_slovenije.pdf (accessed on 5 Dec. 2023)

      It is well known that workers in more physically active occupations and lower education and income (truck drivers, construction workers, etc.) is “hard-to-reach” with digital health interventions, especially male, which is also apply to this research [33, 34].

      1. Rasmussen, C. L.; Dumuid, D.; Hron, K. Day-to-day pattern of work and leisure time physical behaviours: are low socioeconomic status adults couch potatoes or work warriors?. BMC Public Health. 2021, 21, 1342. doi:10.1186/s12889-021-11409-0
      2. Sendall, M. C.; McCosker, L. K.; Crane, P.; Rowland, B.; Fleming, M.; Biggs, H. C. Using Facebook for Health Promotion in "Hard-to-Reach" Truck Drivers: Qualitative Analysis. J Med Internet Res. 2018, 20, e286. doi:10.2196/jmir.9689

Reviewer 2 Report

Comments and Suggestions for Authors

The authors have done a considerable amount of work to correct the manuscript, but nevertheless, I have some comments:

Despite the correction of the title, in my opinion, it does not reflect the essence of the conducted study.

To clarify:

1.                  Engagement Frequency in Web-Based Nutrition and Physical activity Educational Campaign

2.                  Accordingly, the primary objective of this study was to identify the differences in the engagement frequency in e-mail educational campaigns between employees performing different types of work classified as (a) predominantly sedentary, (b) predominantly standing, and (c) predominantly physical, with the additional aim of analyzing these differences by BMI, gender, age and selected indicators of dietary and activity habits in SMEs. We also examined the differences in absenteeism due to NCDs between the SMEs included and those not included in the web-based educational campaigns. (стр.78-84).

3.                  Conclusions

In Slovenia, as in the rest of the world, we are confronted with the problem of obesity and obesity-related diseases. The workplace represents a key focus for public health initiatives aimed at combating obesity in the working population. It is interesting to note that older male sedentary workers with better lifestyle habits were the most engaged participants in our campaigns. This suggests that high-risk workers may not be interested in web-based programs and that gender-specific interventions may prove beneficial. These findings suggest that different approaches may be needed for different workplaces, approaches that are gender and habit specific and promote better motivation and stronger engagement of public health institutions (стр.483-492).

There should be a logical connection between the title, purpose and conclusions. It would have been right for the authors to reflect in the title the importance of work process factors on the degree of involvement in the educational process through e-mail messages.

It is necessary, in my opinion, to prioritize between the way of providing preventive information to respondents, the degree of involvement of employees in the educational process and the factors of the work process affecting this process.

As presented, the interpretation of the obtained results is perceived as illogical, despite the impressive observation period, well-conducted statistical analysis and sufficient quality of the tables and figures.

Author Response

Our responses to each point raised by the Reviewer 2 are written in green font, added/changed text in the article is made by tracking the changes in the text.

Reviewer 2

The authors have done a considerable amount of work to correct the manuscript, but nevertheless, I have some comments:

Despite the correction of the title, in my opinion, it does not reflect the essence of the conducted study.

To clarify:

  1. Engagement Frequency in Web-Based Nutrition and Physical activity Educational Campaign

  1. Accordingly, the primary objective of this study was to identify the differences in the engagement frequency in e-mail educational campaigns between employees performing different types of work classified as (a) predominantly sedentary, (b) predominantly standing, and (c) predominantly physical, with the additional aim of analyzing these differences by BMI, gender, age and selected indicators of dietary and activity habits in SMEs. We also examined the differences in absenteeism due to NCDs between the SMEs included and those not included in the web-based educational campaigns. (стр.78-84).

  1. Conclusions

In Slovenia, as in the rest of the world, we are confronted with the problem of obesity and obesity-related diseases. The workplace represents a key focus for public health initiatives aimed at combating obesity in the working population. It is interesting to note that older male sedentary workers with better lifestyle habits were the most engaged participants in our campaigns. This suggests that high-risk workers may not be interested in web-based programs and that gender-specific interventions may prove beneficial. These findings suggest that different approaches may be needed for different workplaces, approaches that are gender and habit specific and promote better motivation and stronger engagement of public health institutions (стр.483-492).

There should be a logical connection between the title, purpose and conclusions. It would have been right for the authors to reflect in the title the importance of work process factors on the degree of involvement in the educational process through e-mail messages.

It is necessary, in my opinion, to prioritize between the way of providing preventive information to respondents, the degree of involvement of employees in the educational process and the factors of the work process affecting this process.

As presented, the interpretation of the obtained results is perceived as illogical, despite the impressive observation period, well-conducted statistical analysis and sufficient quality of the tables and figures.

Thank you for your thoughtful and considered feedback. To better align the title, purpose, and conclusions—also following the recommendation of Reviewer 1—we initially acknowledged potential challenges in measuring the impact on NCDs. Upon re-examining the data, we found it to be of insufficient quality and usefulness. Consequently, we have excluded all text related to absenteeism due to NCDs from the article. Additionally, we have made corrections to the title, purpose, and conclusions as follows:

  1. Web-based Nutrition and Physical activity Campaigns: Engagement Across Work Types and Employee Characteristics in SMEs
  2. Accordingly, the primary objective of this study was to identify the differences in the engagement frequency in web-based nutrition and physical activity campaigns between employees performing different types of work classified as (a) predominantly sedentary, (b) predominantly standing, and (c) predominantly physical, with the additional aim of analyzing these differences by BMI, gender, age and selected indicators of dietary and activity habits in SMEs.
  3. Conclusions

In Slovenia, as in the rest of the world, we are confronted with the problem of obesity and obesity-related diseases. The workplace represents a key focus for public health initiatives aimed at combating obesity in the working population. Our study delved into the behavioral patterns of 370 employees across 26 SMEs, shedding light on engagement frequencies in web-based educational campaigns, dietary and physical activity habits, and their correlation with demographic factors. Several noteworthy findings emerged, challenging some established notions in the literature. Men with sedentary work demonstrated higher engagement in web-based campaigns, with older employees participating more actively than expected. Healthy dietary habits correlated with increased engagement, emphasizing lifestyle influences. Despite some limitations, our findings contribute valuable insights for workplace health promotion policies and practices. The results underscore the importance of tailoring interventions to specific occupational groups, considering demographic factors, and recognizing the multifaceted influences on employee engagement in health-related initiatives.

Our responses to each point raised by the Reviewer 2 are written in green font, added/changed text in the article is made by tracking the changes in the text.

Reviewer 2

The authors have done a considerable amount of work to correct the manuscript, but nevertheless, I have some comments:

Despite the correction of the title, in my opinion, it does not reflect the essence of the conducted study.

To clarify:

  1. Engagement Frequency in Web-Based Nutrition and Physical activity Educational Campaign

  1. Accordingly, the primary objective of this study was to identify the differences in the engagement frequency in e-mail educational campaigns between employees performing different types of work classified as (a) predominantly sedentary, (b) predominantly standing, and (c) predominantly physical, with the additional aim of analyzing these differences by BMI, gender, age and selected indicators of dietary and activity habits in SMEs. We also examined the differences in absenteeism due to NCDs between the SMEs included and those not included in the web-based educational campaigns. (стр.78-84).

  1. Conclusions

In Slovenia, as in the rest of the world, we are confronted with the problem of obesity and obesity-related diseases. The workplace represents a key focus for public health initiatives aimed at combating obesity in the working population. It is interesting to note that older male sedentary workers with better lifestyle habits were the most engaged participants in our campaigns. This suggests that high-risk workers may not be interested in web-based programs and that gender-specific interventions may prove beneficial. These findings suggest that different approaches may be needed for different workplaces, approaches that are gender and habit specific and promote better motivation and stronger engagement of public health institutions (стр.483-492).

There should be a logical connection between the title, purpose and conclusions. It would have been right for the authors to reflect in the title the importance of work process factors on the degree of involvement in the educational process through e-mail messages.

It is necessary, in my opinion, to prioritize between the way of providing preventive information to respondents, the degree of involvement of employees in the educational process and the factors of the work process affecting this process.

As presented, the interpretation of the obtained results is perceived as illogical, despite the impressive observation period, well-conducted statistical analysis and sufficient quality of the tables and figures.

Thank you for your thoughtful and considered feedback. To better align the title, purpose, and conclusions—also following the recommendation of Reviewer 1—we initially acknowledged potential challenges in measuring the impact on NCDs. Upon re-examining the data, we found it to be of insufficient quality and usefulness. Consequently, we have excluded all text related to absenteeism due to NCDs from the article. Additionally, we have made corrections to the title, purpose, and conclusions as follows:

  1. Web-based Nutrition and Physical activity Campaigns: Engagement Across Work Types and Employee Characteristics in SMEs
  2. Accordingly, the primary objective of this study was to identify the differences in the engagement frequency in web-based nutrition and physical activity campaigns between employees performing different types of work classified as (a) predominantly sedentary, (b) predominantly standing, and (c) predominantly physical, with the additional aim of analyzing these differences by BMI, gender, age and selected indicators of dietary and activity habits in SMEs.
  3. Conclusions

In Slovenia, as in the rest of the world, we are confronted with the problem of obesity and obesity-related diseases. The workplace represents a key focus for public health initiatives aimed at combating obesity in the working population. Our study delved into the behavioral patterns of 370 employees across 26 SMEs, shedding light on engagement frequencies in web-based educational campaigns, dietary and physical activity habits, and their correlation with demographic factors. Several noteworthy findings emerged, challenging some established notions in the literature. Men with sedentary work demonstrated higher engagement in web-based campaigns, with older employees participating more actively than expected. Healthy dietary habits correlated with increased engagement, emphasizing lifestyle influences. Despite some limitations, our findings contribute valuable insights for workplace health promotion policies and practices. The results underscore the importance of tailoring interventions to specific occupational groups, considering demographic factors, and recognizing the multifaceted influences on employee engagement in health-related initiatives.

Round 3

Reviewer 1 Report

Comments and Suggestions for Authors

I appreciate your extensive responses and the revisions made in the manuscript. I have no additional comments.

Author Response

We thank the reviewer for extensive revision and such enrichment of the article.

Kind regards

Reviewer 2 Report

Comments and Suggestions for Authors

I would like to thank the authors for their work in correcting the manuscript, which allowed them to better present the results.

I am satisfied with the authors' responses; however, I suggest that you consider correcting the title to the following interpretation: "Factors influencing the degree of involvement of employees of medium and small enterprises in preventive web-based programs".

Author Response

We thank the reviewer for extensive revision and such enrichment of the article. 

We changed the title according to reviewer s proposal.

Kind regards